# Ultrafast kinetics of the antiferromagnetic-ferromagnetic phase transition in FeRh

G. Li [1✉], R. Medapalli[2,3], J. H. Mentink [1], R. V. Mikhaylovskiy [3], T. G. H. Blank [1], S. K. K. Patel[2], A. K. Zvezdin[4], Th. Rasing[1], E. E. Fullerton [2] & A. V. Kimel [1]

Understanding how fast short-range interactions build up long-range order is one of the most intriguing topics in condensed matter physics. FeRh is a test specimen for studying this problem in magnetism, where the microscopic spin-spin exchange interaction is ultimately responsible for either ferro- or antiferromagnetic macroscopic order. Femtosecond laser excitation can induce ferromagnetism in antiferromagnetic FeRh, but the mechanism and dynamics of this transition are topics of intense debates. Employing double-pump THz emission spectroscopy has enabled us to dramatically increase the temporal detection window of THz emission probes of transient states without sacrificing any loss of resolution or sensitivity. It allows us to study the kinetics of emergent ferromagnetism from the femtosecond up to the nanosecond timescales in FeRh/Pt bilayers. Our results strongly suggest a latency period between the initial pump-excitation and the emission of THz radiation by ferromagnetic nuclei.

[1] Radboud University, Institute for Molecules and Materials, Heyendaalseweg 135, Nijmegen, The Netherlands. [2] Center for Memory and Recording Research, University of California, San Diego, La Jolla, San Diego, CA 92093-0401, USA. [3] Department of Physics, Lancaster University, Bailrigg, Lancaster LA1 4YW, UK. [4] Moscow Institute for Physics and Technology, 9 Institutskiy per., Dolgoprudny, Moscow Region 141701, Russia. ✉email: Qiao.Li@science.ru.nl

The first-order magnetic phase transition in FeRh has attracted considerable attention in material science[1], magnetic recording[2], magnetocalorics[3] and spintronics[4]. Below the phase transition temperature ($T_{PT} = 370$ K), the material is in the antiferromagnetic phase with two antiparallel Fe sublattices while the Rh atoms have no net magnetic moment. Above $T_{PT}$, the material is in the ferromagnetic phase where the Rh atoms gain a magnetic moment and the magnetic moments of the Fe and Rh sublattices align parallel (see Fig. 1a).

The availability of femtosecond laser pulses as ultrafast heat sources initiated a plethora of experimental studies on the speed at which the magnetization emerges. The very first time-resolved experiments showed that when a femtosecond near-infrared pulse excites FeRh, a sub-picosecond magneto-optical Kerr (MOKE) signal assigned to the growth of the net magnetization was observed[5,6]. Later, picosecond X-ray pulses as a probe revealed that the dichroic signal, related to the ferromagnetic order, emerges at the timescale of 100 ps for both the Fe and Rh sublattices[7,8]. A similar timescale of 60 ps for the magnetic response was found while the nucleation of ferromagnetic domains was shown to happen at the same timescale (30 ps) as the lattice dynamics[9]. On the other hand, the electronic structure associated with the ferromagnetic phase reacts at a sub-picosecond timescale[10]. Recently, THz emission from ultrafast magnetization dynamics was reported in FeRh/Pt where THz emission below $T_{PT}$ was suggested to originate from interfacial magnetism[11,12]. Hence the understanding of the kinetics of the femtosecond laser-induced phase transition from the anti-ferromagnetic to the ferromagnetic state in FeRh remains largely unclear and controversial[1,5–7].

A potential reason for this controversy is the fact that practically all time-resolved studies of the emerging ferromagnetism in FeRh have employed pump-probe techniques using only a single pump pulse. The latter require that after each pump pulse the medium relaxes back to the same initial state before the next

pump arrives and the system always follows the same path after excitation. At the same time, the first-order nature of the magnetic phase transition in FeRh implies that ferromagnetic and antiferromagnetic phases can co-exist in a broad temperature range, that can be much broader than the temperature hysteresis may indicate[13] (see Fig. 1a). Several experiments directly demonstrated not only temperature hysteresis, but even ferromagnetic domains nucleating at random positions and temperatures, far below $T_{PT}$[14–19]. In stroboscopic measurements with high pump fluences, after each pump-induced heating and cooling cycle, there is a finite probability that FeRh relaxes either to the initial antiferromagnetic or to the less favorable, but metastable, ferromagnetic state[12,20]. Hence, the next pump pulse can trigger either ferromagnetic order from the antiferromagnetic state or from metastable ferromagnetic domains. In principle, the signals of these two types were not distinguished in single-pump experiments on FeRh so far.

Recently, double-pump THz emission spectroscopy was proposed as a conceptually different approach for time-resolved studies of the magnetic phase first-order transition in FeRh[20]. Although the double-pump THz emission spectroscopy is not immune to artefacts from co-existing phases, we will show that it is better suited for studying the kinetics of first-order phase transitions. In FeRh, were ferromagnetic domains can co-exist in the antiferromagnetic phase, both pumps will launch either ultrafast nucleation of new ferromagnetic domains or demagnetization of residual ferromagnetic domains. By measuring the THz emission as a function of the delay between the two pump pulses, we can deduce the magnetization dynamics of the ferromagnetic phase transition induced by the first pump. Moreover, this technique allows one to detect magnetization dynamics up to nanosecond timescales without sacrificing any sensitivity or temporal resolution. Applying this method to an FeRh/Pt bilayer we get a signal-to-noise ratio of 46 dB in the optimal condition for ultrafast magnetometry of FeRh/Pt and trace the emergence of the laser-induced ferromagnetic order on femto-, pico-, and nanosecond timescales with unprecedented sensitivity. We show that although a femtosecond laser pulse launches ultrafast spin dynamics in the antiferromagnet, the domains generated by this laser pulse do not emit THz waves and are also insusceptible to an external magnetic field during the first 16 ps of their lifetime (see Fig. 1b). It shows that long-range ferromagnetic spin order has not been established yet. In the following these domains will be referred as nuclei of ferromagnetic phase. To support these experimental findings, we propose a phenomenological model according to which this latency is intrinsic to the first-order nature of the phase transition from antiferromagnetic to ferromagnetic states and must be present even in the case when the sign of the exchange interaction changes instantaneously.

## Results

**Single-pump THz emission as a function of temperature.** Epitaxial FeRh thin film was embedded into a MgO/FeRh(40 nm)/Pt(5 nm) heterostructure. The magnetic and structural characterizations confirming the high quality of the studied FeRh film can be found in a ref. [12]. For our single-pump THz emission experiment we applied a 50 fs laser pulse at a central wavelength of 800 nm at a fluence of 13 mJ cm$^{-2}$. An electromagnet was used to apply an in-plane external magnetic field up to 120 mT. Figure 2a shows the traces of the THz electric field emitted under influence of a single-pump excitation. Below $T_{PT}$ ($T = 300$ K) the THz emission is small, but measurable. A temperature increase results in a significant increase and saturation of the signal above $T_{PT}$. The peak amplitude of the THz electric field deduced from the data $E_{Peak}$ is plotted in Fig. 2b as a function of temperature

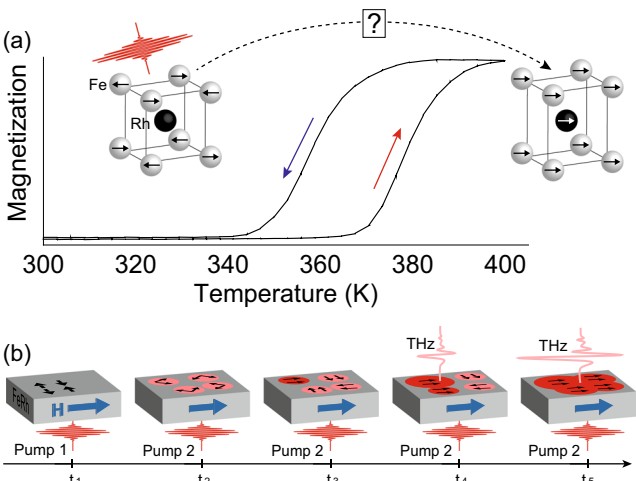

**Fig. 1 First-order phase transition and double-pump THz emission in FeRh. a** The temperature hysteresis of the FeRh magnetization measured by vibrating sample magnetometry[12] shows a characteristic first-order phase transition. **b** The timeline of the laser-induced antiferromagnetic-ferromagnetic phase transition in FeRh under an applied magnetic field $\mu_0 H$ and the principle of the double-pump THz emission spectroscopy. The initial pump (1) launches the phase transition at $t_1$. The subsequent pump (2) arrives at $t_i$ ($i = 2, 3, …$) triggering THz emission from the ferromagnetic nuclei aligned along the magnetic field. The intensity of the THz emission increases with the growing ferromagnetic nuclei with their magnetization along the applied magnetic field.

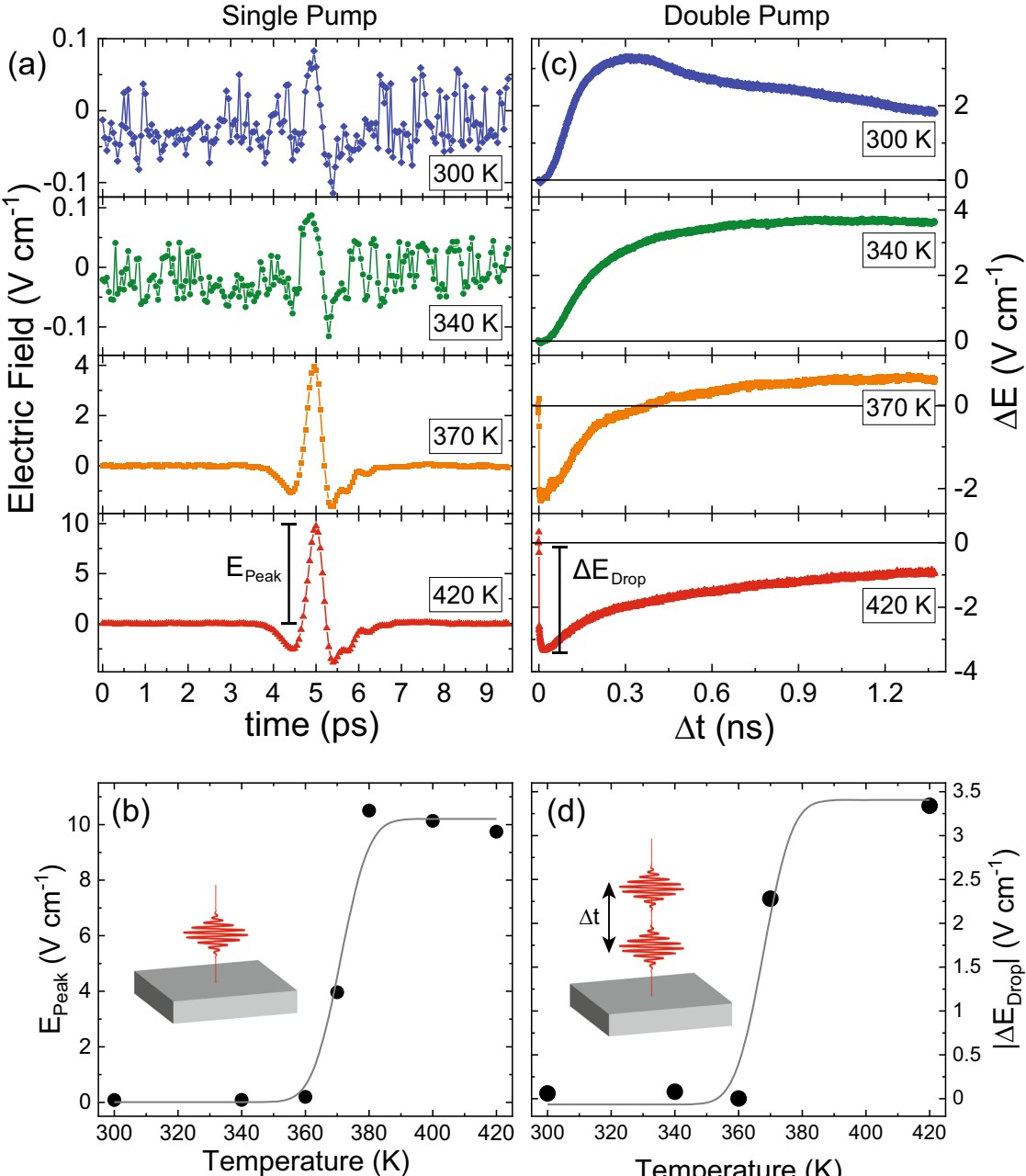

**Fig. 2 Single-pump and double-pump THz emission below and above the temperature of the phase transition $T_{PT}$.** Left column: **a** The time-traces of the THz electric field at several temperatures with $\mu_0H = 120$ mT. **b** The peak amplitude of the THz electric field triggered by a single-pump as a function of initial temperature of the sample. Right column: **c** The change in $\Delta E$ as a function of the delay time, $\Delta t$, between pump 1 and pump 2 at several temperatures with $\mu_0H = 120$ mT. **d** The absolute value of the sub-picosecond drop of the double-pump THz emission, $\Delta E$, as a function of temperature. The solid lines in **b** and **d** serve as a guide to the eye.

and emphasizes that the signal is at maximum in the ferromagnetic phase. A decrease of the sample temperature reduces the volume of the ferromagnetic phase and results in a decrease of the corresponding $E_{Peak}$ signal. Below $T_{PT}$, the ferromagnetic phase is less stable, but thermodynamically allowed[9,14–16,18,19]. In previous works, similar single-pump-induced THz emission was observed where the polarity of the THz signal was the same below and above the phase transition[12,20]. This suggests that the THz emission observed below $T_{PT}$ is of the same origin as the THz emission observed above $T_{PT}$. It is well known that femtosecond laser excitation of a ferromagnet triggers THz emission due to sub-picosecond demagnetization[11,12,21,22]. Therefore, it is natural

to assume that the $E_{Peak}$ signal above $T_{PT}$ is a result of the ultrafast laser-induced demagnetization of ferromagnetic FeRh. Hence the small $E_{Peak}$ signal below $T_{PT}$ can also be assigned to be due to ultrafast demagnetization of the ferromagnetic phase. In order to verify this hypothesis, we employ the technique of double-pump THz emission spectroscopy.

**Double-pump THz emission as a function of temperature.** For this method, we apply two 50 fs laser pulses at a central wavelength of 800 nm. The fluences for pump 1 and 2 were 13 and 10 mJ cm$^{-2}$, respectively. An in-plane external magnetic field up to 120 mT was applied. We detect the effect of the first pump

pulse on the THz electric field emitted under action of the second pump pulse ($\Delta E$). Varying the delay between the pump pulses $\Delta t$ and measuring $\Delta E$ reveals the dynamics of the phase transition between the pulses.

Figure 2c shows the double-pump THz signals $\Delta E$ as a function of time-delay between the two pump-pulses. The sample MgO/FeRh/Pt is pumped from the side of the MgO-substrate and the experiments were performed at various temperatures above and below the phase transition. Below $T_{PT}$, the dynamics of $\Delta E$ shows a slow rise on a timescale of ~300 ps, in agreement with ref. [20]. and similar to the dynamics observed with an X-ray probe[7].

Above $T_{PT}$, $\Delta E$ show strong sub-picosecond dynamics that is opposite in sign with respect to the $\Delta E$ signal measured below $T_{PT}$, followed by a slow relaxation on a nanosecond timescale. Near $T_{PT}$, the dynamics of $\Delta E$ is a combination of the two effects, similar to the results reported in ref. [7]. The amplitude of the sub-picosecond drop $\Delta E_{Drop}$ in the $\Delta E$ signal is shown in Fig. 2d. The behavior of $\Delta E_{Drop}$ is very similar to the one of $E_{Peak}$ in Fig. 2b. Therefore, the sub-picosecond dynamics of the $\Delta E$ signal above $T_{PT}$ can be reliably assigned to the ultrafast demagnetization of the ferromagnetic phase of FeRh. Since the first pump substantially demagnetizes ferromagnetic FeRh, it reduces the ability of the second pump to induce THz emission as a result of ultrafast demagnetization, resulting in a drop in $\Delta E$. The temporal separation of the first and second pump pulses results in a partial recovery of the net magnetization and thus in a partial recovery of the double-pump THz emission.

Below $T_{PT}$, the $\Delta E$ signals have clearly the opposite sign compared to those above $T_{PT}$. This is evidence that the first pump pulse launches a process opposite to the ultrafast demagnetization of a ferromagnetic phase, i.e. the growth of the ferromagnetic phase in FeRh. In particular, the first pump generates ferromagnetic nuclei and triggers the phase transition resulting in an increase of the net magnetization. This enhances the ability

of the second pump to initiate ultrafast demagnetization, resulting in an increase of $\Delta E$.

**Double-pump THz emission as a function of magnetic field.** Figure 3 shows the dynamics of $\Delta E$ of FeRh at 300 K with an in-plane applied magnetic field. The $\Delta E$ signal shows a clear dependence to the external magnetic field $\mu_0 H$ and a small, but observable signal is seen even in zero field with a preferred direction for the growth. This dependence to the magnetic field confirms that the $\Delta E$ signal is a measure of the net magnetization $\Delta M$ of ferromagnetic nuclei generated by pump 1. The preferred direction of the magnetization growth in zero field can be weakly defined by parasitic fields including those generated by residual ferromagnetic domains that may exist even far below $T_{PT}$[12,23]. Moreover, Fig. S2 in supplementary(2) reveals a fast dynamics of the $\Delta E$ signal at the first picoseconds which is opposite in sign with respect to the slow dynamics. This fast dynamics is similar to the ultrafast TR-MOKE signal reported in refs. [5,6]. However, the opposite sign with respect to the slow dynamics signal suggests that the fast dynamics originates from the demagnetization of the residual domains.

**Electric- and magnetic dipole sources of THz emission.** Figure 4 again shows the double-pump THz signals $\Delta E$ as a function of time-delay between two pump-pulses $\Delta t$. This time, the measurements were performed at various temperatures and for the cases when the femtosecond pulses are incident from the side of the Pt-film and from the side of the MgO-substrate, respectively. In our experiments the pump was incident either from the Pt or MgO face of the sample and the detected THz signal was emitted from the MgO or Pt face, respectively. We switched between these two configurations by rotating the sample by 180° around the axis of the in-plane magnetic field. It is clear that the dynamics in these two cases is substantially different. The large difference in

**Fig. 3 Dynamics of the double-pump THz emission $\Delta E$ measured at 300 K. a** The $\Delta E$ signal for various external magnetic field strengths and **b** for two opposite polarities at $\mu_0 H = 120$ mT.

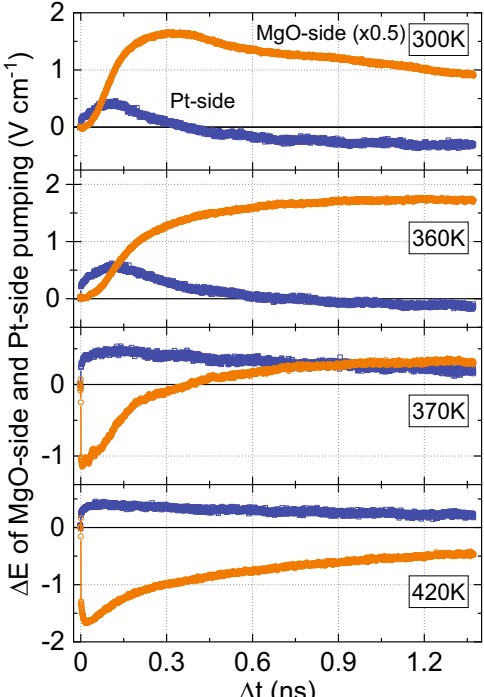

**Fig. 4 MgO-side and Pt-side pumped THz emission.** Dynamics of the double-pump THz emission $\Delta E$, at $\mu_0 H = 120$ mT for the cases when the MgO/FeRh/Pt heterostructure is pumped from the MgO-side (orange) or the Pt-side (blue).

the $\Delta E$ signal when pumped from the MgO-side or the Pt-side shows that the THz emission originates from at least two sources with different symmetries with respect to space inversion. The two sources of THz emission interfere constructively when pumped from the MgO-side, resulting in a larger $\Delta E$ signal and that they interfere destructively when pumped from the Pt-side, resulting in a smaller $\Delta E$ signal and even changing sign below $T_{PT}$.

The mechanisms for generating THz emission from these two sources are explained in the methods section. The weakest source, which is invariant under space inversion, is of magnetic dipole origin. Indeed, the THz electric field that is emitted by the rapidly changing magnetization in the bulk of FeRh does not change sign upon rotating the sample around the applied field. The strongest source is of electric dipole origin[21,22] and the corresponding THz electric field does change sign upon changing the pumping side of the sample.

This fundamental difference in the behavior of the sources under space inversion can be employed to retrieve the dynamics of the magnetic ($E^{MD}$) and electric ($E^{ED}$) dipole contributions to the laser-induced changes in the THz emission.

The dipole contributions to the $\Delta E$ signal pumped from the MgO-side are expressed as $\Delta E^{MgO} = E^{MD} + E^{ED}$ and from the Pt-side the signal is $\Delta E^{Pt} = E^{MD} - E^{ED}$. Hence in order to obtain the best signal-to-noise ratio, one has to perform the measurements pumping from the MgO-side, when the two sources interfere constructively.

**Dynamics of the laser-induced emergence of the net magnetization.** In order to suppress the THz emission from the residual ferromagnetic domains below $T_{PT}$ (300 K) as well as contributions of non-magnetic origin[24], we employed the fact that the dynamics of the newly laser-induced and randomly oriented ferromagnetic nuclei in FeRh must be susceptible to an applied magnetic field. At the (sub-)picosecond timescale, the law of conservation of angular momentum dictates that the speed of the magnetization dynamics $\left(\frac{d\mathbf{M}}{dt}\right)$ scales with the magnetic field ($\mu_0\mathbf{H}$) as $\frac{d\mathbf{M}}{dt} = -\gamma \mathbf{M} \times \mu_0\mathbf{H}$, where $\gamma/2\pi = 28$ GHz/T is the gyromagnetic ratio. Therefore, picosecond magnetization dynamics can only occur in effective magnetic fields of the order of 10 T and similarly strong applied fields are needed to affect such dynamics. At external magnetic fields much less than 10 T, the dynamics of sub-picosecond demagnetization of ferro- and ferrimagnetic metals do not depend on the magnetic field strength[25–27]. This should also be the case for the ferromagnetic nuclei in FeRh.

The single-pump THz emission performed on the sample, reported in ref. ([12], N. Fig. 4) showed a magnetic hysteresis above $T_{PT}$ corresponding to the MOKE signal in Fig. S3 (see supplementary(3)) where the magnetization saturates at 15 mT. Hence, if any residual ferromagnetic domains are present in the predominantly antiferromagnetic phase and a femtosecond laser pulse causes their sub-picosecond demagnetization, the latter should not be affected by fields above 15 mT. Therefore, in order to remove the part of the signal originating from sub-picosecond demagnetization of the residual ferromagnetic domains as well as signals from non-magnetic sources, we took the data measured at 15 mT as a baseline and subtracted it from measurements obtained at higher field strengths.

The $\Delta E$ signal in Fig. 5 was normalized (explained in supplementary(4)) such that it represents the ferromagnetic fraction. We found that it was insufficient to fit our data with the sum of an exponential rise and decay function to extract the rise and relaxation times, respectively, as was done in ref. [7]. The large mismatch especially below 100 ps suggests that there must be an

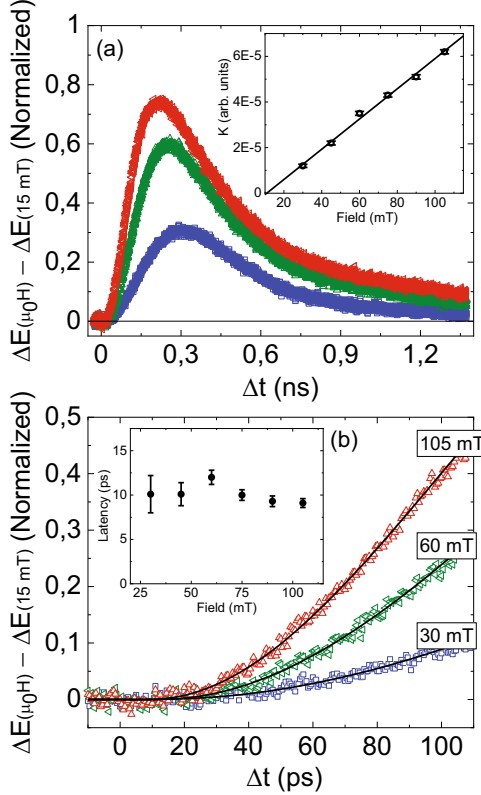

**Fig. 5 The double-pump THz emission due to emerging ferromagnetism in antiferromagnetic FeRh/Pt, $\Delta E(\mu_0 H)$–$\Delta E(15$ mT) ($\mu_0 H = 30$, 60, and 105 mT) at 300 K. a** shows the sub-nanosecond dynamics. The inset shows the extracted K coefficient as a function of the applied field. The linear line is a guide to the eye. **b** shows the picosecond dynamics. The solid lines are the fits to Eq. (1). The inset shows the extracted latency as a function of the applied field.

additional regime where there is a latency in the dynamics of $\Delta E$. We therefore fitted the $\Delta E$ signal with a function that was inspired by the Johnson-Mehl-Avrami-Kolmogorov equation (explained in supplementary(5)) (ref. [28], pp. 16–21)[29,30],

$$f(t) = \mathcal{H}\left(t - t_{latency}\right)\left(1 - e^{-K(t - t_{latency})^2}\right). \quad (1)$$

Here $\mathcal{H}(t)$ is the Heaviside-step-function and $t_{latency}$ is the latency where before $t_{latency}$ there is no $\Delta E$ signal. After $t_{latency}$, the growth of the $\Delta E$ signal is determined by $K$. Figure 5b shows the fit based on Eq. (1) with the data for three different field strengths below 100 ps. The inset of Fig. 5a shows that the growth rate, $K$, is linearly proportional to the magnetic field strength which suggests that $K$ describes the growth of the ferromagnetic phase. The estimated latency as a function of applied field is shown in the inset of Fig. 5b).

**THz emission from MgO/FeRh(40 nm)/Au(5 nm).** It is interesting to compare our data with those on double-pump THz emission from a bare FeRh sample reported earlier. Although a direct comparison is hampered by the lack of calibration of the THz signals in ref. [20], here we compare MgO/FeRh/Pt with MgO/FeRh/Au. Figure 6 shows the single-pump THz emission signal in MgO/FeRh/Pt and MgO/FeRh/Au at 400 K. The THz electric field traces are shown for pumping from the MgO- and Pt(Au)-side, respectively. In the case of MgO/FeRh/Pt, the THz signal changes sign under space inversion (see Fig. 6a). This shows that the electric dipole contribution to the THz emission is larger than

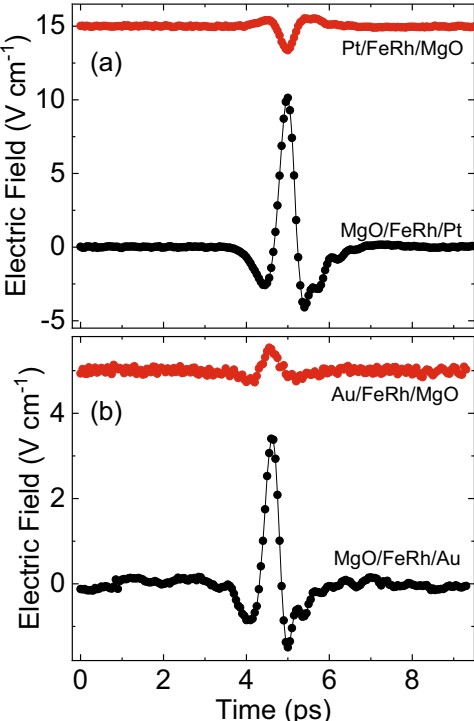

**Fig. 6 Single pump THz emission from MgO/FeRh/Pt and MgO/FeRh/Au.** Comparison of the single-pump THz emission in MgO/FeRh/Pt (**a**) and MgO/FeRh/Au (**b**). The measurements were performed at 400 K and $\mu_0 H$ = 120 mT. pumping either from the MgO- (black) or Pt(or Au)-(red) side.

the magnetic dipole contribution. For MgO/FeRh/Au, the THz signal does not change sign under space inversion (see Fig. 6b). This suggests that the electric dipole contribution to the THz emission is weaker than the magnetic dipole contribution. Indeed, the spin-Hall angle of Au is an order of magnitude smaller than that of Pt which will result in a weaker electric dipolar THz emission[31].

We further performed double-pump THz emission experiments on this sample and deduced the signal originating from the emerging ferromagnetism in antiferromagnetic FeRh (see Fig. 7), $\Delta E(\mu_0 H) - \Delta E(30\,mT)$, as explained above. Since the FeRh/Pt and FeRh/Au interfaces must result in different anisotropies[32,33] and magnetization dynamics also depends on magnetic anisotropy for both the antiferromagnetic and ferromagnetic phases, it would be interesting to see how the kinetics of the magnetic phase transition is affected by differences in capping layer. The latency for FeRh/Au, where the THz emission is dominated by magnetic dipole sources, is shown in the inset of Fig. 7b. The error margins are much higher compared to FeRh/Pt due to the low signal-to-noise ratio. While the data do not contradict the possible presence of a latency, the experimental uncertainty due to the low signal-to-noise ratio hampers quantitative estimates of the latter.

## Discussion

Our double-pump results show a latency between the arrival of the initial pump and the rise of the net magnetization. This cannot be observed with single-pump THz emission since it cannot probe dynamics slower than 3 ps. Nonetheless, a THz signal was observed below $T_{PT}$ in the single-pump THz emission experiment. This is due to the fact that any single-pump-probe experiment cannot distinguish between a signal from residual ferromagnetic domains or laser-induced nuclei. As explained in ref. [12], the sub-picosecond magnetization dynamics in FeRh reported earlier[5,6] can be explained by ultrafast demagnetization

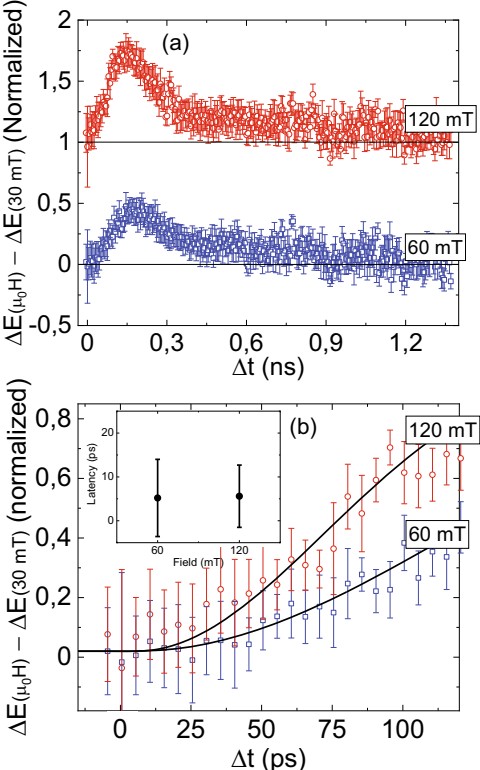

**Fig. 7 The double-pump THz emission signals due to emerging ferromagnetism in antiferromagnetic FeRh/Au, $\Delta E(\mu_0 H) - \Delta E(30\,mT)$ ($\mu_0 H$ = 60, and 105 mT) at 300 K.** The data are binned for clarity. Panel **a** shows sub-nanosecond dynamics and panel **b** shows the picosecond dynamics with the fit to Eq. (1). The inset shows the latency as a function of applied field.

of residual ferromagnetic domains. Moreover, the observed latency in the ability of the laser-induced nuclei to emit THz radiation is in agreement with the findings of ref. [9], that reports that the phase transition from the antiferromagnetic to the ferromagnetic state has two intrinsic timescales. First, the initial nucleation of the ferromagnetic nuclei, which is the same for both magnetic and structural dynamics. Second, for the subsequent growth and alignment of the ferromagnetic nuclei to the applied magnetic field. The characteristic time of the structural changes, $\tau$, is defined by the speed of sound $v$ and the film thickness $h$ ($\tau = h/v$). However, the processes that evolve in the spin system at this timescale have not been discussed yet and remain unclear.

To understand the origin of the observed latency in response to the laser pulses, we first consider a system of two classical spins $\mathbf{m}_1$ and $\mathbf{m}_2$ coupled by an exchange interaction such that the energy is $E = -J\mathbf{m}_1 \cdot \mathbf{m}_2$. For $J < 0$ the spins are aligned anti-parallel in the antiferromagnetic ground state and for $J > 0$ the spins are aligned parallel in the ferromagnetic state. Even if $J$ changes sign instantaneously, dynamics of these spins can only be induced by fluctuations, since in the collinear situation there is no net torque. We propose that $J$ can be changed by strain effects which will drive the dynamics and the fluctuations and cause the reorientation of spins in random directions. Therefore, the response of the system features a multi-domain state with a latency between the change of the sign of $J$ and the emergence of a net magnetization. We do not consider laser-induced quenching of the antiferromagnetic order to play a significant role in the laser-induced phase transition. A complete quenching of the spin order is highly unlikely since the quenching of the ferromagnetic order is only 40%[7].

To gain a qualitative understanding, we consider a phenomenological and deterministic model of two macroscopic spins with magnetizations $\mathbf{M}_1$ and $\mathbf{M}_2$, respectively. Considering a homogeneous heating across FeRh, the free energy is given by:

$$F = f\left(M_1^2\right) + f\left(M_2^2\right) - J_{\text{eff}}\mathbf{M}_1 \cdot \mathbf{M}_2 - H_0 \cdot \left(\mathbf{M}_1 + \mathbf{M}_2\right) + \frac{1}{2}\epsilon u^2. \tag{2}$$

Here $f\left(M_i^2\right)$ determines the non-equilibrium exchange energy of sublattice $i$ ($i = 1, 2$)[34], $J_{\text{eff}} \equiv J_{\text{FeFe}}^{(1)} + \rho_J u + J_{\text{FeFe}}^{(2)}\mathbf{M}_1 \cdot \mathbf{M}_2$ is the isotropic Heisenberg exchange, with a distance dependence parametrized by $\rho_J = (\partial J_{\text{FeFe}}^{(1)}/\partial u)$ the magneto-structural constant from Kittel[35], $J_{\text{FeFe}}^{(2)}$ is the effective four-spin exchange parameter[36], $\mathbf{H}_0$ is the external magnetic field along the x-axis and $\epsilon$ is the elastic stiffness constant describing the structural contribution. The equilibrium condition $\left(\partial F/\partial u\right)_T = 0$ implies $\langle u \rangle = \rho_J(\mathbf{M}_1 \cdot \mathbf{M}_2)/\epsilon$ yielding an effective 4-spin contribution of the form $F_{\text{eq}}^{(2)} = -(J_{\text{FeFe}}^{(2)} + \rho_J^2/2\epsilon)(\mathbf{M}_1 \cdot \mathbf{M}_2)^2$, which demonstrates that the lattice-dependent exchange term exhibits the same symmetry as the effective four-spin interactions $J_{\text{FeFe}}^{(2)}(\mathbf{M}_1 \cdot \mathbf{M}_2)^2$ derived in refs. [36,37]. In equilibrium, this model shows a first-order phase transition from the AFM to the FM phase, caused by the competition between the isotropic Heisenberg exchange and the 4-spin contribution, in accordance with atomistic simulations for FeRh[37] and analogous to the first-order transition in AFM CuO[38,39]. To model the kinetics of the AFM-FM transition, we consider both longitudinal and transverse dynamics determined by the equations[40]:

$$\frac{1}{\gamma}\frac{d\mathbf{M}_{1,2}}{dt} = \mu\lambda_{\text{so}}\mathbf{H}_{1,2} + \mu\lambda_{\text{ex}}(\mathbf{H}_{1,2} - \mathbf{H}_{2,1}) + \mathbf{M}_{1,2} \times \mathbf{H}_{1,2}, \tag{3}$$

where the effective fields $\mathbf{H}_{1,2} \equiv -\partial F/\partial\mathbf{M}_{1,2}$ are derived from the free energy[34]:

$$\mathbf{H}_{1,2} = -\frac{k_{\text{B}}T}{\mu_{\text{Fe}}}\mathcal{L}^{-1}\left(\frac{|\mathbf{M}_{1,2}|}{\mu_{\text{Fe}}}\right)\frac{\mathbf{M}_{1,2}}{|\mathbf{M}_{1,2}|} + J_{\text{eff}}\mathbf{M}_{2,1} + \mathbf{H}_0. \tag{4}$$

The relativistic constant $\lambda_{\text{so}}$ describes angular momentum transfer to the lattice, the exchange-related $\lambda_{\text{ex}}$ allows for exchange-mediated angular momentum transfer between the sublattices, $\gamma$ is the gyromagnetic ratio of iron, $\mu = g\mu_{\text{B}}$ with $g$ the g-factor of iron and $\mathcal{L}(x)$ is the Langevin function. Note that the first term on the right-hand side of (5) can be written in first-order approximation as $-A(\mathbf{M}_{1,2}^2 - \mathbf{M}_0^2)\mathbf{M}_{1,2}$ where $A > 0$ is a constant and $\mathbf{M}_0$ the saturation magnetization at the given temperature. We investigate the response of this system of equations to a change of sign of the effective parameter $J_{\text{eff}}(u(t), t)$, for which the timescale depends on the exact microscopic origin of $J_{\text{eff}}$, which out of equilibrium may differ from the equilibrium case. For generality, Fig. 8 shows the simulations both for an instantaneous change of sign and for the case when the timescale is set by that of the fastest possible structural phase transition (~10 ps), limited by the speed of sound[41]. Both experimental and computational results reveal a latency period in the growth of net magnetization. Fitting the simulated data for the case when the exchange constant $J_{\text{eff}}$ changes sign instantaneously at $\mu_0 H = 105$ mT with our modified Johnson-Mehl-Avrami-Kolmogorov equation, we found a latency of $9.33 \pm 0.01$ ps, which is in good agreement with the experimentally observed latency. If the effective exchange parameter $J_{\text{eff}}$ gradually changes within 30 ps[9], the latency window also extends, practically following $J_{\text{eff}}$.

Note that the slope $d\mathbf{M}(t)/dt$ after the latency is indeed different in the simulations compared with the experiment, but this is certainly not surprising. In our simple two-spin model, after the sign change of the exchange interaction and after the latency,

$d\mathbf{M}(t)/dt$ is defined by the strength of the effective field of the exchange interaction. This is of the order of $10^2$–$10^3$ Tesla. In real samples, the appearance of ferromagnetic nuclei with emerging magnetizations in random directions makes the dynamics more complex. A more realistic model of FeRh should therefore consist of many of these antiparallel spins, which after a latency period will nucleate into many ferromagnetic domains that will subsequently grow. However, such a computational study, including moving domain walls, is presently a challenge and certainly a complete project of its own. Despite the oversimplification, a latency period before the growth of magnetization is present nonetheless. This shows that this period right after the laser-excitation cannot be ignored.

## Summary

In summary, our results show a latency between the laser-induced heating of FeRh and the emergence of a net magnetization at its antiferromagnetic to ferromagnetic phase transition. Our phenomenological model of the magneto-structural phase transition between the antiferromagnetic and ferromagnetic phase in FeRh shows that the latency must be a fundamental property of the phase transition. In particular, the latency of the emergent ferromagnetic order should show up even under an instantaneous change of the effective exchange interactions. Therefore, our work strongly suggests that, within the framework of our simple model, the latency, previously identified in ref. [9], must be a general feature of first-order magnetic phase transitions and not necessarily unique to FeRh or even to a particular FeRh film. At the same time, it is also clear that more work would need to be done on the model for FeRh and other similar systems to establish true generality. We expect that our results will inspire new theoretical studies of spin dynamics with time-dependent exchange interactions beyond adiabatic approximations as well as motivate the development of novel experimental techniques allowing to reveal the ultrafast dynamics of the exchange interaction itself.

## Methods

**Sample fabrication.** For our experiments we fabricated epitaxial FeRh thin films embedded into MgO/FeRh(40 nm)/Pt(5 nm) and MgO/FeRh(40 nm)/Au(5 nm) heterostructures. The samples were grown on MgO(001) substrates via sputter deposition. The FeRh layer was deposited at 450 °C and was annealed at 800 °C afterwards for 45 min. The Pt(Au) capping layer was deposited afterwards at room temperature. The X-ray reflection and diffraction measurements confirm the high quality of FeRh(001) films and the magnetic phase transition was characterized via vibrating sample magnetometry[12].

**THz emission spectroscopy.** Time-resolved THz emission spectroscopy is a well-established experimental technique for studying magnetization dynamics and spintronic phenomena at picosecond and sub-picosecond timescales[21,42–45]. If a femtosecond laser pulse causes a sub-picosecond change of the magnetization or launches a sub-picosecond photocurrent pulse in the sample plane, the sample will emit THz radiation of magnetic dipole or electric dipole origin, respectively. Previous studies have shown that the THz emission from a magnetic dipole source in ferromagnets can be directly linked to ultrafast magnetization dynamics, where the electric field of the emitted radiation is proportional to a current source, $\mathbf{E} \sim \mathbf{J} = \nabla \times \mathbf{M}$[42,46].

In magnetic/non-magnetic heavy metal bilayers, such as FeRh/Pt, spin currents originating from laser-induced ultrafast demagnetization propagate across the interface and are converted, in Pt, to charge currents due to the inverse spin-Hall effect, $\mathbf{j}_c \sim \mathbf{M} \times \mathbf{j}_s$[21,22]. In the simplest approximation Ohm's law, $\mathbf{E} = \sigma^{-1}\mathbf{j}_c$, implies that an electric field is emitted by a varying electric dipole (time-varying current). Hence, the strength of the THz emission is a measure of ultrafast laser-induced magnetization dynamics with both magnetic and electric dipole sources.

However, the typical time-resolved THz emission spectroscopy setup is hardly sensitive to electromagnetic waves at frequencies below 100 GHz. This makes the detection of spin dynamics slower than 10 ps virtually impossible.

**Double-pump THz emission spectroscopy.** The idea of our double-pump experiment is to generate nuclei of the ferromagnetic phase with the initial first femtosecond pump pulse (pump 1). As soon as the nuclei become ferromagnetic, i.e. acquire a net magnetization, their excitation with the subsequent second femtosecond pump pulse (pump 2) will launch sub-picosecond demagnetization

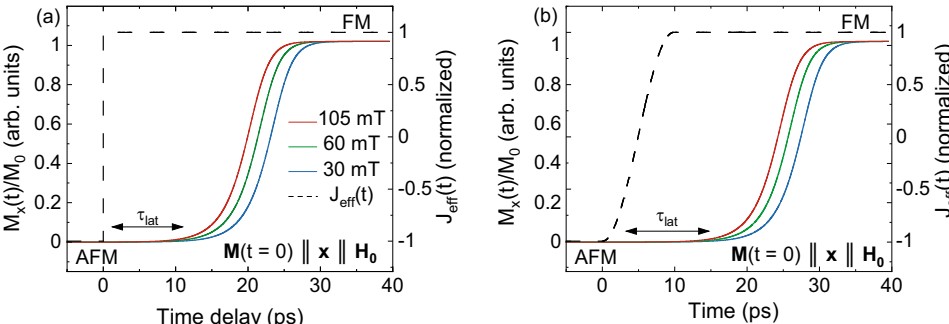

**Fig. 8 Simulated net magnetization M(t) using a model that mimics the behavior of FeRh using 2 macrospins.** The simulations assume that the magnetic parameters are $|J_{eff}| = 24.8$ meV/$\mu^2$, $M_0 = 0.1\mu$, $A = 4|J_{eff}|$, $\lambda_{ex} = 0.025$, $\lambda_{so} = 0.2\lambda_{ex}$ and $g = 2$[6,37,48]. The system starts in the collinear antiferromagnetic (AFM) phase with spins along the x-axis (analogous results are obtained for other initial orientations). The dashed and solid lines indicate the normalized exchange constant ($J_{eff}$) and magnetization, respectively. In **a** $J_{eff}$ changes sign instantaneously while for **b** it changes at the time-scale of the lattice-expansion. In either case, after demagnetization of the antiferromagnetic vector, ferromagnetic (FM) order along the $\hat{x}$-axis grows only after a latency period $\tau_{lat} \approx 10$ ps. The speed at which ferromagnetic order emerges at $\tau_{lat}$ depends on the magnitude of the external magnetic field $|\mu_0 \mathbf{H}_0| = 30, 60, 105$ mT.

resulting in THz emission. Measuring how the first pump pulse affects the THz emission triggered by the second pump pulse, will be a measure of the magnetization induced in the medium by the first pump. Varying the delay between the pump pulses we are able to reveal the emergence of the laser-induced magnetization, as illustrated in Fig. 1b.

The double-pump technique uses two linearly polarized optical pump pulses. The schematic of the experimental setup is shown in supplementary(1). The first pump pulse (pump 1) excites the medium at a repetition rate of 500 Hz, while the repetition rate of the second pump (pump 2) is at 1 kHz. Both pump pulses are weakly focused down to a 2-mm beamspot at the sample to maximize the signal-to-noise ratio. The pump fluence dependence in Fig. S7 of supplementary(7) shows that the fluence of pump 2 does not affect the ferromagnetic response while the fluence dependence of pump 1 drives the magnetic phase transition similarly as reported in ref. [7]. The duration of the pulses is 50 fs and the fluence is 13 and 10 mJ cm$^{-2}$ for pump 1 and pump 2, respectively. A variable external magnetic field $\mu_0$H was applied to control the direction of the in-plane magnetization of the sample. The emitted THz radiation is collimated and focused onto a ZnTe crystal by two gold plated parabolic mirrors. Using electro-optical sampling with the help of the ZnTe crystal[47] and a gate pulse at a time delay $t$, we detect the electric field of the emitted THz radiation under action of the second pump. The time delay $t$ is fixed at the position where the THz peak amplitude under action of the second pump overlaps in time with the gate pulse. The delay stage of the first pump is used to vary the time delay ($\Delta t$) between both pumps. If pump 2 does not overlap with pump 1 and arrives later, using lock-in detection at the reference frequency defined by the repetition rate of pump 1, we will detect only that part of the THz emission from pump 2, which was induced by pump 1 ($\Delta E$). By swapping the reference frequency to the repetition rate of pump 2, the lock-in will detect the THz emission strength from pump 2. A detailed discussion on the double-pump lock-in detection method is found in supplementary(1). Day-to-day fluctuations in the setup resulted in the differences in the detected $\Delta E$. This resulted in a lower $\Delta E$ peak signal in Fig. 3b while having a high applied magnetic field compared to the $\Delta E$ signal in Fig. 3a.

## Data availability

All data presented in this study are available from the corresponding authors upon reasonable request.

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

## Acknowledgements

This work was funded by the DOE grant #DE-SC0003678, the Netherlands Organization for Scientific Research (NWO), the European Union Horizon 2020 and innovation program under the FET-Open grand agreement no.713481 (SPICE) and the European Union's Horizon 2020 research and innovation program under the Marie Skłodowska-Curie grant agreement No 861300 (COMRAD), the European Research Council ERC grant agreement no.856538 (3D-MAGiC) and ERC grant agreement no.852050 (MAGSHAKE), and the Shell-NWO/FOM-initiative 'Computational sciences for energy research' of Shell and Chemical Sciences, Earth and Life Sciences, Physical Sciences, FOM and STW. A.K.Z. acknowledges the support from the Russian Science Foundation under grant No. 20-42-08002. We want to thank Sergey Semin, and Chris Berkhout from Radboud University, and Ray Descoteaux from CMRR, UCSD for technical support.

## Author contributions

R.V.M., A.V.K., and E.E.F. designed the research. G.L. and R.M. performed the experiments. G.L. performed the data analysis. S.K.K.P. fabricated the samples. J.H.M., T.G.H.B., A.K.Z., and A.V.K. developed the theory. G.L., R.M., R.V.M., J.H.M., T.G.H.B., and A.V.K. wrote the paper with important contributions from A.K.Z., Th.R., and E.E.F. The project was coordinated by A.V.K.

## Competing interests

The authors declare no competing interests.
