## [Peer Review File · Nature Communications]

Reviewers' Comments:

Reviewer #1:

Remarks to the Author:

In this work the authors present a mostly experimental study of the laser-driven AFM-FM transition in FeRh, a system that has been discussed extensively in the ultrafast magnetism community. The authors apply a relatively recently developed experimental technique, what they call "double pump" THz emission. This is discussed and the data analysed in the context of a phenomenological model of the phase transition. They make several main claims in the conclusions: (1) That there is a "latency" period of 20 +/- 5 ps where the nucleation centres of new ferromagnetic domains are not susceptible to an external field, and (2) this latency time is a fundamental property of all first order magnetic phase transitions.

The general scientific questions addressed in this work are relevant, and indeed this method may yield useful insights. The authors I think overstate the controversy on this material; although early tr-MOKE measurements (ref. 4 and 5) did indeed make the surprising claim of an "ultrafast" FM phase, the subsequent work has showed that this was an artefact of the measurement technique (cf. refs. 7 and 8, the latter of which actually redoes the tr-MOKE and shows the lack of an ultrafast component). Nonetheless, it is true that the dynamics of the magnetic phase transition are not well understood and that FeRh is an excellent model system to study this.

This said, I am not convinced that the authors adequately support their main claims, and indeed their claims appear to be contradicted by existing literature and so I am not prepared to recommend this for publication in Nature Communications. In addition, there are several major apparent inconsistencies that prevent me at present from recommending this paper for publication in general. Below I list my concerns.

1) One major issue that I have is the apparent claim from the authors that performing measurements with the pump coming through the backside of the sample (through the substrate) is equivalent to performing the measurements upon spatial inversion. This is just not true, and I don't see any reasonable argument for claiming that it is. Spatial inversion would require inverting everything except the applied H field. I'm not sure why the authors didn't just use an inverted applied H field to try to separate magnetic dipole origins from electric dipole origins. The method that the authors use instead is not correct and will give misleading results, since there are pretty easy to see differences between pumping through the Pt layer vs the substrate due to the depth dependence of the laser absorption. Thus I doubt that much of the discussion around Fig 4 is very meaningful.

2) The authors claim that "residual nuclei" of the FM phase are insusceptible to an applied field < 1 T (see lines 201-202 and 206-207), but I don't see any support for this statement in ref. 25, which is not even about FeRh. Thus I am uncertain about the validity of the analysis procedure which is based on this claim.

3) The claim that there is no ferromagnetic signal at all for the first 20 ps is not well supported. First, as I discussed in point 2 the procedure of subtracting the 15 mT data is a bit questionable. Second, the authors do not show the data without this subtraction in a clear plot for the time period in question, which makes it difficult to assess if indeed the data have sufficient signal-to-noise to really see if anything unusual happens at 20 ps. All data have limited signal-to-noise, and it is therefore not surprising that for a slow onset effect like this that there is a period of time when it is hard to distinguish a signal from noise. That is not the same thing as a "latency" time. In fact, as I mention below (point 8), in the XMCD data of ref. 7 there is a small but measurable signal in this time frame, which would seem to argue against the authors' main claim. Also, the authors do not give enough information about the constraints used for the linear fits in Fig 5b.

4) The authors claim that the double pump measurements show that the small signal observed in the "single pump" THz emission is from residual FM domains that are metastable at low temperatures, but I do not see any support for this. In fact, after line 93 I do not see any mention of this claim again at all.

5) The overall claim that earlier tr-MOKE measurements are seeing dynamics of the residual ferromagnetic nuclei (lines 269-271) is not well supported. Simply not seeing an effect in this experiment is not proof that there is nothing to see. Also, the authors do not seem to give a plausible explanation of how these residual FM nuclei could result in the tr-MOKE signals observed.

6) The phenomenological model is not sufficiently well explained and justified. Why do the authors assume that the lattice expands on the same time scale as the lattice temperature increase, when they themselves pointed out that this expansion happens on a time scale given by the film thickness divided by the speed of sound? I also don't see how it is reasonable to propose a model that appears to assume a homogeneous magnetic state, when for a first order transition it is clear that nucleation plays a critical role.

7) On line 104, the authors use ref 7 as an example of previous work that also shows that the double pump change in emission field changes on a time scale of 300 ps. However, this reference does not report THz emission at all, it is a X-ray MOKE study. I get that the magnetic dynamics inferred from the measurement are similar, but the way the reference is worded the reader is led to believe that this is a THz emission study.

8) The authors misstate the overall conclusions from Ref. 7. This work does not show that there are no changes in the XMCD signal of FeRh up to 10 ps. This number of 10 ps was the time resolution of some of the experimental data, but the data in Fig. 3 do indeed show that in the overlap region between the pump and probe for the 10 ps duration probe pulse there is a measurable XMCD change.

9) In the introduction the authors discuss the limitations of stroboscopic measurements, which is reasonable in principle but they also seem to imply that their measurements are somehow exempt from the problems presented. This is misleading, in my opinion. Despite the authors' claims that the "double pump" method is fundamentally different, in fact it is really just a different type of stroboscopic pump-probe measurement where the second "pump" is really a probe. I also don't see how the authors' experiments cannot be influenced by hysteresis effects.

Minor comments, which can be optionally addressed:

10) The authors should give at least a brief explanation of the parameters of the single pump and double pump measurements in the main text, before the discussion of results. It is not clear for many of the data important details like the orientation and strength of the applied H field, or the fluences of the pulses. I know some of this is given in the Methods section, but it should be possible to follow the paper without flipping back and forth.

11) Overall I would suggest that the authors minimize the interpretation of the data when first discussing the results. The argumentative style of writing when new data are presented is distracting and makes it difficult to assess what is directly from the measurement and what is an interpretive argument from the authors.

Reviewer #2:

Remarks to the Author:

The authors of this manuscript provided a detailed study of the AFM-to-FM phase transition of FeRh utilizing a double-pump THz emission setup. By variation of the sample temperature, external magnetic field, and excitation geometry they found a significant dependence of the THz emission strength on the presence of the FM phase in FeRh. The authors verified their findings by comparing to FeRh sample with Pt and Au cap layer. The authors found two regimes of the photo-driven phase transition: 1. a 10-20 ps delay of the emergence of the FM phase after photoexcitation, which might be a "fundamental property" of the phase transition and 2. a magnetic field dependence of the growth velocity of FM domains lasting a few 100 ps. The experimental findings were partially resembled by a macro-spin model accounting for magneto-structural interactions.

Since I have several technical/methodical as well as physical concerns with the current form of the manuscript, I cannot recommend publication without major changes and clarifications.

General remarks:

1. I am a bit confused by the term nuclei which is used by the authors to describe the nucleation of FM domains. I would suggest either explaining the current meaning of this term in their context or changing the wording to FM domain nucleation instead.
2. In my view there can be two pathways from the AFM towards the FM phase: either "directly" by realigning the AFM spins to local FM order which then grows to larger domains. I guess this is what happens in the temperature-driven phase transition. So in essence there is always local magnetic order present. But as a second route, one could completely destroy any magnetic order starting from the AFM phase using a fs-laser pulse and then recover from above the Curie temperature of the FM phase. Do the authors have any arguments for or against one or the other scenario? Please remind yourself, that the excitation in the sample will not be homogeneous, neither in- or out-of-plane because of the Gaussian beam profile and exponential absorption profile.
3. The authors state that the THz double pump emission spectroscopy has the advantages of a more defined relaxation of the sample state, compared to conventional pump-probe approaches. I actually do not see that! The authors also do a typical pump-and-probe experiment at 1kHz rep. rate and the sample must relax back to an initial and defined state. The only difference is that their probe is generated by another pump. In my opinion, this only leads to a double-excited state which could result in a more defined final state of the dynamics, in case of reaching some kind of saturation of the dynamics, e.g. the Curie temperature of Fe. But the same could be also achieved by strong enough pump pulses in e.g. MOKE setups. On the other hand, I agree that the THz emission spectroscopy is a rather sensitive method to probe FM order, which, however, can also have non-magnetic contributions. So I don't see the point why this experiment could not be done with MOKE or Faraday setups? Could the authors even carry out conventional MOKE measurements with their setup?
4. The authors give an overview of the controversial previous results on the phase transition of FeRh, which also triggered their study. However, I miss a thorough discussion of the discrepancies between the different studies and the results of this study. I can imagine that different techniques probe different sample volumes and that the sample geometry might also have an influence on the dynamics.
5. In the last experimental part, the authors subtract low-field data (15 mT) in order to get rid of residual FM and non-magnetic contributions. Shouldn't it be possible to use zero-field data here, as both contributions are discussed to be not influenced by external magnetic fields? Or alternatively, why didn't the authors toggle the external field and use half of the difference signal as extracted transient magnetic signal? Again the two former contributions should not depend on low magnetic fields below 1 T.
6. The authors compared front- and back-side pumping of the samples and noted that reflection and absorption effects can be neglected. Is there any measurement or estimation for this assumption? In the case of Pt-side pumping, I expect a significant deposition of energy in the Pt layer compared to back-side pumping. To that end, the magnetization dynamics should have very different amplitudes and the THz emission from a hot or cold Pt layer should be also very different as its conductivity changes.
7. I am not very convinced by the theoretical model, which the authors present. They state that it results in a delay of about 10 ps even for an instantaneous change of the exchange interactions. The change of the exchange interaction is linked to the expansion of the lattice, which the authors correctly link to the sound velocity of the FeRh layer (L279). Later in L320 they discuss a sub-picosecond timescale of the change of the exchange interaction due to an ultrafast heat-induced lattice expansion. But there is no sub-ps lattice expansion in a 40 nm thick film! There can only be quasi-instantaneous STRESS but not STRAIN! Based on this fact the change of the exchange interaction will take also several ps and the authors cannot distinguish this effect from any

“fundamental” delay in the phase transition. In order to do so, the authors must study the FeRh thickness or probing-depth dependence of this delay or have an additional structural probe at hand.

To this end, I am convinced that the proposed model is not correct or simply assumes way too few atomic layers (which of course need much less time to expand than a 40 nm thick film).

Accordingly, there might be no fundamental delay of the phase transition, but it is simply limited by the structural phase transition and further delayed by the growth of the FM domains.

8. I was also wondering if the authors can give any estimate on the actual delay, between the max. of the emitted THz pulse and the actual pump-pulse. I think this delay is already reasonable and at least in the range of 10 % of the proposed fundamental delay – so it must be discussed here as well.

9. What is the FM saturation field strength to obtain a single-domain state above the phase transition temperature? Is there an estimate about the domain sizes below the saturation field?

10. Why would the fundamental delay of the phase transition be so different in the Pt vs. the Au sample? I think there is a significant difference between the two experimentally obtained values? The presented theory does not take the cap layer into account at all.

Specific comments:

L81: replace “dramatic” with a more meaningful/quantitative adjective, like “100 times” or “significant”

L95-96: Both pump pulses initiate magnetization dynamics in a broader sense. The authors might differentiate between AFM and FM magnetization dynamics or need to be more specific.

L118: again the authors should carefully distinguish between FM and AFM “magnetization” as also the AFM phase has a sub-lattice magnetization.

L132: please describe the “directions” of the magnetic field here.

Figure 3: The electric field strength in (b) at 120 mT is smaller than in (a) for 105 mT. This is not consistent and must be explained.

Figure 4: again the field strength in (a) 300 K is different than in Figure 3.

Line 179-80: $dE(t=x)$ will be zero at the point where dE changes sign. Hence Eq. (1) will be multiplied by zero!?

L215 & Figure 5: The authors state that their fit function results in a large mismatch with the experimental data. This is obvious, as they do not include the delay of the exponential rise. Please fit the data with a double exponential fit function that includes a delay. Otherwise the inset in Fig. 5 (a) is also rather meaningless. Please also scale the inset to maximize the view on the presented data.

In (b) one should not fit a linear function in order to determine the delay of the onset. Please use again a step- or error-function in combination with a linear function instead.

Figure 7: please bin the data and add error bars instead of cluttering the complete plot with data points.

Again the fits in (a) and (b) do not look very convincing to me.

Figure 9: Can one neglect heat diffusion out of the phononic system of FeRh into the Pt cap and MgO substrate layer? The FM saturation is reached already after 30 ps in the simulation, while the experiment needs about 300 ps, can you discuss the discrepancy? Why does the FM saturation moment not depend on the external magnetic field as in the experiment. Moreover it is only 0.8 μB and not the full 3.8 μB of Fe. I think the 3.8 μB for Fe is also a rather large value.

Please also shift the time zero from 3 ps to 0 ps in accordance to the experiment. Accordingly, the

onset at 10 ps is only 7 ps after the first pump pulse in the simulation and not approx. 10 ps!
I would also appreciate the usage of experimental SI units even for the simulations.

L 321: heath = heat

L395: the second pump pulse should be much smaller than the first one, in order to probe a homogeneously excited region with no in-plane gradients.

L402-03: this assumes that the delay, phase, and spectrum of the THz pulses do not change throughout the delay scan. Have the authors checked this by carrying out full EOS spectra at different pump1-pump2 delays?

REVIEWER COMMENTS

Reviewer #1 (Remarks to the Author):

In this work the authors present a mostly experimental study of the laser-driven AFM-FM transition in FeRh, a system that has been discussed extensively in the ultrafast magnetism community. The authors apply a relatively recently developed experimental technique, what they call “double pump” THz emission. This is discussed and the data analysed in the context of a phenomenological model of the phase transition. They make several main claims in the conclusions: (1) That there is a “latency” period of 20 +/- 5 ps where the nucleation centres of new ferromagnetic domains are not susceptible to an external field, and (2) this latency time is a fundamental property of all first order magnetic phase transitions.

The general scientific questions addressed in this work are relevant, and indeed this method may yield useful insights. The authors I think overstate the controversy on this material; although early tr-MOKE measurements (ref. 4 and 5) did indeed make the surprising claim of an “ultrafast” FM phase, the subsequent work has showed that this was an artefact of the measurement technique (cf. refs. 7 and 8, the latter of which actually redoes the tr-MOKE and shows the lack of an ultrafast component). Nonetheless, it is true that the dynamics of the magnetic phase transition are not well understood and that FeRh is an excellent model system to study this.

This said, I am not convinced that the authors adequately support their main claims, and indeed their claims appear to be contradicted by existing literature and so I am not prepared to recommend this for publication in Nature Communications. In addition, there are several major apparent inconsistencies that prevent me at present from recommending this paper for publication in general. Below I list my concerns.

0) We thank the Referee for the time invested in reviewing our manuscript. The feedback helped us to significantly improve our manuscript. However, we respectfully disagree with the Referee in his statement about contradictions between our claims and existing literature. We would like to address these points below.

1) One major issue that I have is the apparent claim from the authors that performing measurements with the pump coming through the backside of the sample (through the substrate) is equivalent to performing the measurements upon spatial inversion. This is just not true, and I don't see any reasonable argument for claiming that it is. Spatial inversion would require inverting everything except the applied H field. I'm not sure why the authors didn't just use an inverted applied H field to try to separate magnetic dipole origins from electric dipole origins. The method that the authors use instead is not correct and will give misleading results, since there are pretty easy to see differences between pumping through the Pt layer vs the substrate due to the dept dependence of

the laser absorption. Thus I doubt that much of the discussion around Fig 4 is very meaningful.

1) We apologize for not being clear with our descriptions in the manuscript. Our goal is indeed to separate the THz emission of magnetic dipole and electric dipole sources, which rely on different physical mechanisms. While the former originates directly from the laser-induced ultrafast decrease of the magnetization of ferromagnetic FeRh (*i.e.* demagnetization), the latter is an indirect result of the demagnetization which, in turn, leads to spin injection into the Pt-layer and subsequent conversion of this spin-current into charge current due to the spin Hall effect ([T. Kampfrath *et al.*, *Nature Nanotech* **8**, 256–260 (2013)] and [O Gueckstock *et al.*, *Adv. Mater.* **33**, 2006281 (2021)]). This charge current is responsible for the THz emission of electric dipole origin. As both sources originate from demagnetization, however, inverting the direction of the magnetic field will invert the sign of them both. We would like to emphasize that the method applied to our work is widely used in the analysis of THz spectroscopy ([W. Zhang *et al.*, *Nature Commun* **11**, 4247 (2020)] and [T. Huisman *et al.*, *Nature Nanotech* **11**, 455–458 (2016)] and [T. Seifert *et al.*, *Nature Photon* **10**, 483–488 (2016)] and [E. Papaioannou *et al.*, *Nanophotonics*, **10** (4), pp. 1243-1257 (2020)] and [Y. Wu *et al.*, *Adv. Mater.* **29**, 1603031 (2017)] and [Qi Zhang *et al.*, *Phys. Rev. Applied* **12**, 054027 (2019)]). The discussion is therefore anything but meaningless.

2) The authors claim that “residual nuclei” of the FM phase are insusceptible to an applied field < 1 T (see lines 201-202 and 206-207), but I don’t see any support for this statement in ref. 25, which is not even about FeRh. Thus I am uncertain about the validity of the analysis procedure which is based on this claim.

2) We apologize for this misunderstanding. In fact, the sentence was meant to make a very general statement which is valid to all magnetic metals including FeRh. In particular, we would like to state that changing the orientation of a single electron spin must obey the fundamental law of conservation of angular momentum $\frac{dS}{dt} = -\gamma \mathbf{S} \times \mu_0 \mathbf{H}$, where $\gamma = 28 \text{ GHz/T}$ is the gyromagnetic ratio. Hence the characteristic strength of magnetic fields involved in the process of sub-picosecond demagnetization is of the order of 100 T (*i.e.* the characteristic fields of the exchange interaction) and magnetic fields of 1 T are simply unable to affect sub-picosecond magnetization dynamics.

3) The claim that there is no ferromagnetic signal at all for the first 20 ps is not well supported. First, as I discussed in point 2 the procedure of subtracting the 15 mT data is a bit questionable. Second, the authors do not show the data without this subtraction in a clear plot for the time period in question, which makes it difficult to assess if indeed the data have sufficient signal-to-noise to really see if anything unusual happens at 20 ps. All data have limited signal-to-noise, and it is therefore not surprising that for a slow onset effect like this that there is a period of time when it is hard to distinguish a signal from noise. That is not the same thing as a “latency” time. In fact, as I mention below (point 8), in the XMCD data of ref. 7 there is a small but measurable signal in this time

frame, which would seem to argue against the authors' main claim. Also, the authors do not give enough information about the constraints used for the linear fits in Fig 5b.

3) We respectfully disagree with the Referee and are afraid that the Referee misunderstood us. Our statement of “no ferromagnetic signal” was based on the fact, that ferromagnetic nuclei in an external magnetic field must satisfy the fundamental law of conservation of angular momentum $\frac{d\mathbf{M}}{dt} = -\gamma\mathbf{M} \times \mu_0\mathbf{H}$, *i.e.* the rate at which the magnetization changes depends linearly on the applied magnetic field strength. Figure 5 shows that such a dependence is seen only after 20 ps. As Ref. [I. Radu *et al.*, *Phys. Rev. B* **81** 104415 (2010)] does not compare measurements at different magnetic fields in the range 0-20 ps, it simply cannot contradict our finding. The linear fit shown in Fig. 5(b) is fitted to the data points measured between 20-40 ps. We have added an additional line to clarify this constraint of the fit.

4) The authors claim that the double pump measurements show that the small signal observed in the “single pump” THz emission is from residual FM domains that are metastable at low temperatures, but I do not see any support for this. In fact, after line 93 I do not see any mention of this claim again at all.

4) We apologize for not being more clear. The statement has been also published earlier [R. Medapalli *et al.*, *Applied Physics Letters* **117**, 142406 (2020)] and we therefore thought it acquired less attention in this paper. This conclusion can be drawn from single pump THz emission experiments, where the shape and polarity of the signal is the same below and above the phase transition. Since above the phase transition, the only possible effect of femtosecond laser excitation on magnetism of a metal is sub-picosecond demagnetization, one has to accept that also below the temperature of the phase transition the signal must be caused by ultrafast demagnetization as well.

We clarified this statement and added the reference to the previous work.

5) The overall claim that earlier tr-MOKE measurements are seeing dynamics of the residual ferromagnetic nuclei (lines 269-271) is not well supported. Simply not seeing an effect in this experiment is not proof that there is nothing to see. Also, the authors do not seem to give a plausible explanation of how these residual FM nuclei could result in the tr-MOKE signals observed.

5) Thank you for pointing this out, so we can be more clear in our claims. In fact, one of the goals of our manuscript is to convince readers that the only possible process that can be active in FeRh on the sub-picosecond time scale is ultrafast demagnetization of ferromagnetic nuclei. The statement is not limited to tr-MOKE measurements only and valid for all techniques sensitive to the net magnetization.

Note that any single-pump stroboscopic measurement (not only tr-MOKE) cannot distinguish whether the signal comes from pre-existing nuclei or laser-induced nuclei.

This problem can be avoided by using a double pump experiment to separate the signal from pre-existing and newly formed nuclei.

We added these additional arguments in our revised manuscript to strengthen our claims. See lines 112-118

6) The phenomenological model is not sufficiently well explained and justified. Why do the authors assume that the lattice expands on the same time scale as the lattice temperature increase, when they themselves pointed out that this expansion happens on a time scale given by the film thickness divided by the speed of sound? I also don't see how it is reasonable to propose a model that appears to assume a homogeneous magnetic state, when for a first order transition it is clear that nucleation plays a critical role.

6) We understand the concern of the referee and apologize for the confusion. We do not claim that we have built a complete model of FeRh, but in fact we simply show that the kinetics of a system comprising essentially two macroscopic spins, where the exchange interaction is suddenly changed, can successfully reproduce the observed latency. It strongly suggests that this latency must be a fundamental property of first-order magnetic phase transitions.

We agree with the referee that it may be confusing to link the lattice expansion to both lattice temperature and speed of sound. Therefore, in the revised version we have simplified the model. Taking the time-dependency of J_{eff} as a free parameter, we focus on two regimes: an instantaneous change of J_{eff} and a change in sync with the lattice expansion. In both cases the model shows a latency, further supporting that the latency is a fundamental property of the phase transition.

7) On line 104, the authors use ref 7 as an example of previous work that also shows that the double pump change in emission field changes on a time scale of 300 ps. However, this reference does not report THz emission at all, it is a X-ray MOKE study. I get that the magnetic dynamics inferred from the measurement are similar, but the way the reference is worded the reader is led to believe that this is a THz emission study.

7) We thank the Referee for pointing this out and have corrected this in the revised version.

8) The authors misstate the overall conclusions from Ref. 7. This work does not show that there are no changes in the XMCD signal of FeRh up to 10 ps. This number of 10 ps was the time resolution of some of the experimental data, but the data in Fig. 3 do indeed show that in the overlap region between the pump and probe for the 10 ps duration probe pulse there is a measurable XMCD change.

8) We thank the Referee for pointing this out and have revised our statements.

9) In the introduction the authors discuss the limitations of stroboscopic measurements, which is reasonable in principle but they also seem to imply that their measurements are somehow exempt from the problems presented. This is misleading, in my opinion. Despite the authors' claims that the "double pump" method is fundamentally different, in fact it is really just a different type of stroboscopic pump-probe measurement where the second "pump" is really a probe. I also don't see how the authors' experiments cannot be influenced by hysteresis effects.

9) We apologize for this confusion. We strongly disagree with the Referee that our method is in fact single-pump, double probe method. The fundamental difference with a conventional pump-probe experiment is that here we use not two, but three pulses which define the measured signal where the two pump pulses have different repetition rates. More particularly, the first pump generates nuclei that evolve in the following time domain towards a ferromagnetic state, the second pump pulse demagnetizes these ferromagnetic nuclei and causes their THz emission which is read out by the third (probe) pulse using the principle of electro-optical sampling. In this way, our detection technique is focused on the nuclei generated by the first pulse and can follow their evolution over time.

The important difference is that the single pump technique will collect signals from both the demagnetization of the pre-existing ferromagnetic domains as well as the emerging magnetization of the nuclei generated by this pump. Our technique allows one to distinguish signals from the pre-existing and newly generated ferromagnetic nuclei. In this respect, co-existing ferromagnetic and antiferromagnetic phases and, consequently, temperature hysteresis do not hamper our measurements.

Minor comments, which can be optionally addressed:

10) The authors should give at least a brief explanation of the parameters of the single pump and double pump measurements in the main text, before the discussion of results. It is not clear for many of the data important details like the orientation and strength of the applied H field, or the fluences of the pulses. I know some of this is given in the Methods section, but it should be possible to follow the paper without flipping back and forth.

10) We thank the Referee for the suggestions. We have made the proper changes.

11) Overall I would suggest that the authors minimize the interpretation of the data when first discussing the results. The argumentative style of writing when new data are presented is distracting and makes it difficult to assess what is directly from the measurement and what is an interpretive argument from the authors.

11) We thank the Referee for this suggestion and made additional changes.

Reviewer #2 (Remarks to the Author):

The authors of this manuscript provided a detailed study of the AFM-to-FM phase transition of FeRh utilizing a double-pump THz emission setup. By variation of the sample temperature, external magnetic field, and excitation geometry they found a significant dependence of the THz emission strength on the presence of the FM phase in FeRh. The authors verified their findings by comparing to FeRh sample with Pt and Au cap layer. The authors found two regimes of the photo-driven phase transition: 1. a 10-20 ps delay of the emergence of the FM phase after photoexcitation, which might be a “fundamental property” of the phase transition and 2. a magnetic field dependence of the growth velocity of FM domains lasting a few 100 ps. The experimental findings were partially resembled by a macro-spin model accounting for magneto-structural interactions.

Since I have several technical/methodical as well as physical concerns with the current form of the manuscript, I cannot recommend publication without major changes and clarifications.

0) We thank the Referee for critically examining our manuscript. The feedback has helped us to improve our manuscript greatly. We will address the Referees comments point-by-point below.

General remarks:

1. I am a bit confused by the term nuclei which is used by the authors to describe the nucleation of FM domains. I would suggest either explaining the current meaning of this term in their context or changing the wording to FM domain nucleation instead.

1) We thank the Referee for this valuable suggestion. We have added additional clarification on this term at the introduction of our manuscript, as proposed by the Referee.

2. In my view there can be two pathways from the AFM towards the FM phase: either “directly” by realigning the AFM spins to local FM order which then grows to larger domains. I guess this is what happens in the temperature-driven phase transition. So in essence there is always local magnetic order present. But as a second route, one could completely destroy any magnetic order starting from the AFM phase using a fs-laser pulse and then recover from above the Curie temperature of the FM phase. Do the authors have any arguments for or against one or the other scenario? Please remind yourself, that the excitation in the sample will not be homogeneous, neither in- or out-of-plane because of the Gaussian beam profile and exponential absorption profile.

2) Thank you for this insightful question. Our interpretation is that the laser pulse induces a strain on the lattice resulting in a local change in sign of the effective exchange interaction. The antiferromagnetically ordered spins then realign locally to a

ferromagnetic order which then grows with time. In the proposed model, magnetic order is never completely destroyed. However, strictly speaking the arguments for one or the other scenario mentioned by the Referee can only be found by detecting the ultrafast dynamics of the antiferromagnetic Neel vector. We are not aware of such measurements in FeRh.

3. The authors state that the THz double pump emission spectroscopy has the advantages of a more defined relaxation of the sample state, compared to conventional pump-probe approaches. I actually do not see that! The authors also do a typical pump-and-probe experiment at 1kHz rep. rate and the sample must relax back to an initial and defined state. The only difference is that their probe is generated by another pump. In my opinion, this only leads to a double-excited state which could result in a more defined final state of the dynamics, in case of reaching some kind of saturation of the dynamics, e.g. the Curie temperature of Fe. But the same could be also achieved by strong enough pump pulses in e.g. MOKE setups. On the other hand, I agree that the THz emission spectroscopy is a rather sensitive method to probe FM order, which, however, can also have non-magnetic contributions. So I don't see the point why this experiment could not be done with MOKE or Faraday setups? Could the authors even carry out conventional MOKE measurements with their setup?

3) We apologize for not clarifying this confusion. The fundamental difference with a conventional pump-probe experiment is that here we use not two, but three laser pulses where the two pump pulses differ in repetition frequency. In our case, the first pump generates nuclei of ferromagnetic domains that evolve in the following time domain towards a ferromagnetic state, the second pump pulse demagnetizes these ferromagnetic nuclei which causes THz emission that is read out by the third (probe) pulse using the principle of electro-optical sampling. In this way, our detection technique yields a study of the ferromagnetic nuclei generated by the first pulse and follows their evolution in time.

The important difference is that a single pump-probe technique will collect signals from both the ferromagnetic domains, which already existed before the arrival of the pump, and the ferromagnetic nuclei generated by this pump. In contrast, our technique allows to distinguish signals from the pre-existing and newly generated ferromagnetic nuclei. This cannot be achieved by any conventional MOKE pump-probe setups without adding an additional pump.

4. The authors give an overview of the controversial previous results on the phase transition of FeRh, which also triggered their study. However, I miss a thorough discussion of the discrepancies between the different studies and the results of this study. I can imagine that different techniques probe different sample volumes and that the sample geometry might also have an influence on the dynamics.

4) Thank you. We have added further discussion on the different studies.

5. In the last experimental part, the authors subtract low-field data (15 mT) in order to get rid of residual FM and non-magnetic contributions. Shouldn't it be possible to use zero-field data here, as both contributions are discussed to be not influenced by external magnetic fields?

Or alternatively, why didn't the authors toggle the external field and use half of the difference signal as extracted transient magnetic signal? Again the two former contributions should not depend on low magnetic fields below 1 T.

5) Toggling the external magnetic field and using half of the difference signal is indeed an alternative way to extract the transient magnetic signal. However, this method would also include the signal coming from the ultrafast demagnetization of pre-existing ferromagnetic domains since this is also magnetic in origin. Therefore, in order to remove any magnetic signal coming from the pre-existing domains, we subtract the signal at the magnetic field strength of 15 mT since this saturates all pre-existing domains along the magnetic field. Another way to remove the pre-existing domains is to work at lower temperatures which gets rid of pre-existing ferromagnetic domains.

6. The authors compared front- and back-side pumping of the samples and noted that reflection and absorption effects can be neglected. Is there any measurement or estimation for this assumption? In the case of Pt-side pumping, I expect a significant deposition of energy in the Pt layer compared to back-side pumping. To that end, the magnetization dynamics should have very different amplitudes and the THz emission from a hot or cold Pt layer should be also very different as its conductivity changes.

6) We have reported the THz emission on the front- and back-side pumping in the supplementary material of ref. [Medapalli *et al.*, *Appl. Phys. Lett.*, **117** (14), 142406 (2020)]. The THz emission signal at 300K showed no significant difference in the THz signal strength between the front- and back-side pumping. We have added these arguments in the revised manuscript.

7. I am not very convinced by the theoretical model, which the authors present. They state that it results in a delay of about 10 ps even for an instantaneous change of the exchange interactions. The change of the exchange interaction is linked to the expansion of the lattice, which the authors correctly link to the sound velocity of the FeRh layer (L279). Later in L320 they discuss a sub-picosecond timescale of the change of the exchange interaction due to an ultrafast heat-induced lattice expansion. But there is no sub-ps lattice expansion in a 40 nm thick film! There can only be quasi-instantaneous STRESS but not STRAIN! Based on this fact the change of the exchange interaction will take also several ps and the authors cannot distinguish this effect from any "fundamental" delay in the phase transition. In order to do so, the authors must study the FeRh thickness or probing-depth dependence of this delay or have an additional structural probe at hand.

To this end, I am convinced that the proposed model is not correct or simply assumes

way too few atomic layers (which of course need much less time to expand than a 40 nm thick film).

Accordingly, there might be no fundamental delay of the phase transition, but it is simply limited by the structural phase transition and further delayed by the growth of the FM domains.

7) We apologize for the confusion. The “sub-picosecond timescale” of the lattice expansion is a misprint, it should be picoseconds and we thank the referee for pointing that out.

We do not claim that we have built a complete model of FeRh, we simply show that a two spin system, where the exchange interaction is suddenly changed, can successfully reproduce the observed behaviour of the latency. To make this point more clear, we have simplified the model where we will show the magnetization dynamics for two situations: (i) the exchange interaction changes instantaneously and (ii) the exchange interaction changes at the timescale of the speed of sound. This is done to distinguish the fundamental delay in the phase transition from the change in the exchange interaction due to lattice expansion. We have estimated the characteristic time of the structural changes based on the speed of sound reported in [D. W. Cooke *et al.*, *Phys. Rev. Lett.* **109**, 255901 (2012)]. This characteristic time (~10 ps) also corresponds to the lattice dynamics reported in literature. In both cases a latency is observed after the change of the exchange interaction. This strongly suggests that this latency must be a fundamental property of a first-order magnetic phase transition.

8. I was also wondering if the authors can give any estimate on the actual delay, between the max. of the emitted THz pulse and the actual pump-pulse. I think this delay is already reasonable and at least in the range of 10 % of the proposed fundamental delay – so it must be discussed here as well.

8) In our double pump THz emission experiment we detect the THz emission triggered by pump 2 while the change of the THz electric field amplitude is induced by pump 1 (the actual pump pulse). In our setup the time-overlap between the THz peak amplitude triggered by pump 2 and the probe pulse is fixed and we scan by varying the delay of pump 1. Therefore, the response of the change in the THz peak amplitude is the same as the temporal resolution of our setup which is about 50 fs.

9. What is the FM saturation field strength to obtain a single-domain state above the phase transition temperature? Is there an estimate about the domain sizes below the saturation field?

9) The ferromagnetic saturation is 15 mT. We have measured the magnetic hysteresis curve of FeRh in the ferromagnetic phase at 400 K and added to in the supplementary file. We cannot make any estimation of the domain sizes in our sample, but according to Ref. [C Baldasseroni *et al.*, *J. Phys.: Condens. Matter* **27**, 256001 (2015)] the domain size in similar samples is of the order of 1 μm .

10. Why would the fundamental delay of the phase transition be so different in the Pt vs. the Au sample? I think there is a significant difference between the two experimentally obtained values? The presented theory does not take the cap layer into account at all.

10) To understand the origin of the difference in the latency for FeRh/Au and FeRh/Pt, one has to be able to experimentally distinguish exchange-driven spin dynamics from the dynamics of the exchange interaction. However, this is not yet possible and the difference remains unclear to us. Moreover, our model only takes into account the spin systems and lattice expansion of FeRh. We did not include any capping layers.

Specific comments:

L81: replace “dramatic” with a more meaningful/quantitative adjective, like “100 times” or “significant”

We revised the word to “significant”.

L95-96: Both pump pulses initiate magnetization dynamics in a broader sense. The authors might differentiate between AFM and FM magnetization dynamics or need to be more specific.

We revised it to “If the first pump generates ferromagnetic nuclei, then the second pump initiates the ultrafast demagnetization of these nuclei, which results in THz emission.”

L118: again the authors should carefully distinguish between FM and AFM “magnetization” as also the AFM phase has a sub-lattice magnetization.

We revised the paragraph and clarified that the first pump launches a process opposite to the ultrafast demagnetization of the ferromagnetic phase *i.e.* the growth of the ferromagnetic phase in FeRh.

L132: please describe the “directions” of the magnetic field here.

We added additional explanation that an in-plane magnetic field was applied.

Figure 3: The electric field strength in (b) at 120 mT is smaller than in (a) for 105 mT. This is not consistent and must be explained.

The two datasets of Fig. 3(a) and 3(b) were taken weeks apart. Small day-to-day fluctuations in the alignment in the setup resulted in a lower ΔE peak at the highest magnetic field strength.

Figure 4: again the field strength in (a) 300 K is different than in Figure 3.

The same as with Fig 3, fluctuations in the alignment resulted in a lowered field strength. Some dataset were taken weeks to months apart.

Line 179-80: $dE(t=x)$ will be zero at the point where dE changes sign. Hence Eq. (1) will be multiplied by zero!?

We apologize for the confusion, $\Delta E(t=x)$ is the MgO-side pumped ΔE signal. This is not zero. We take the value of the MgO-side pumped ΔE signal at the timestamp $t=x$ where the Pt-side pumped THz signal is zero. We have clarified our equation in our revised manuscript.

L215 & Figure 5: The authors state that their fit function results in a large mismatch with the experimental data. This is obvious, as they do not include the delay of the exponential rise. Please fit the data with a double exponential fit function that includes a delay. Otherwise the inset in Fig. 5 (a) is also rather meaningless. Please also scale the inset to maximize the view on the presented data.

In (b) one should not fit a linear function in order to determine the delay of the onset. Please use again a step- or error-function in combination with a linear function instead.

We want to emphasize that the fit was first done with the understanding that there were only two regimes of dynamics, the rise and relaxation. The mechanism of the latency is still not understood well enough to simply fit it with some modified function. Modifying the function without a good physical reason would also change the other extracted fit parameters that will put doubt in the overall explanation of the fit.

We removed the overall fit of the data in Fig. 5(a) and instead added this as a supplementary figure.

Figure 7: please bin the data and add error bars instead of cluttering the complete plot with data points.

Again the fits in (a) and (b) do not look very convincing to me.

We have binned the data in Fig. 7 and removed the fits in Fig. 7(a).

Figure 9: Can one neglect heat diffusion out of the phonoic system of FeRh into the Pt cap and MgO substrate layer? The FM saturation is reached already after 30 ps in the simulation, while the experiment needs about 300 ps, can you discuss the discrepancy? Why does the FM saturation moment not depend on the external magnetic field as in the experiment. Moreover it is only 0.8 μB and not the full 3.8 μB of Fe. I think the 3.8 μB for Fe is also a rather large value.

Please also shift the time zero from 3 ps to 0 ps in accordance to the experiment. Accordingly, the onset at 10 ps is only 7 ps after the first pump pulse in the simulation and not approx. 10 ps!

I would also appreciate the usage of experimental SI units even for the simulations.

We focused in our model on the qualitative description of the transition to the ferromagnetic state, in particular the latency behavior. Therefore, we did not include any heat diffusion processes. Moreover, we consider a homogeneous heating across the whole sample, which is justified by the fact that the film is only 40 nm. This results in the ferromagnetic saturation and discrepancy in the timescale seen in the simulation. In our revised version, we have simplified our model and normalized the saturation moment of Fe for clarity.

L 321: heath = heat

This misspelling has been corrected

L395: the second pump pulse should be much smaller than the first one, in order to probe a homogeneously excited region with no in-plane gradients.

We chose to keep the beamspot equally large to maximize the signal-to-noise ratio. Moreover, a smaller spot of second pump pulse did not affect the overall dynamics.

L402-03: this assumes that the delay, phase, and spectrum of the THz pulses do not change throughout the delay scan. Have the authors checked this by carrying out full EOS spectra at different pump1-pump2 delays?

We have provided below the THz electric field trace at two time-delays of the two pumps. The right graph shows that the THz waveform does not change throughout the delay scan.

Reviewers' Comments:

Reviewer #1:

Remarks to the Author:

In their revision, the authors have partly addressed and responded to some of the points raised earlier. While some issues are clearer, there are still several major outstanding issues with the current version that preclude me from recommending it. I discuss the points using the same numbering as in my original report.

1) A significant part of the argumentation of the paper relies upon the idea that it is possible to separate magnetic dipole and electric dipole contributions of the THz emission by performing two measurements, one where the sample is pumped from the Pt side and the other where it is pumped from the MgO side. After looking through the references provided by the authors that claim this is a "standard practice," I suspect that one issue may be an imprecise description of the method. In each of those references, the issue of whether THz emission is predominantly from demagnetization itself or from the spin-hall effect is answered by performing measurements where the sample is rotated by 180 degrees about the applied in-plane H field. Is that what the authors actually did here? It is not clear. If I read the paper literally, the procedure is that the *only* thing that changes is to realign the pump so that it pumps the sample from the MgO side, but the sample itself and the detection are unchanged. This is clearly not adequate for differentiating magnetic dipole (MD) from electric dipole (ED) contributions, since what really needs to change is the sample orientation relative to the detector. The authors would at a minimum need to clarify this issue.

Assuming that what is really happening here is that the sample is rotated by 180 degrees (flipped), it is still an open question if this really makes it possible to quantitatively differentiate MD and ED contributions. This makes a number of other assumptions that are not clearly stated and also not clearly justified. One such assumption is that the two pump pulses excite the sample with the same spatial profile for both geometries. Can the authors justify this? I suspect it might be difficult, since ~ 20 nm is the approximate attenuation length of the pump in FeRh. In reviewing the papers cited by the authors as previous applications of their method, I did not find any that tried to quantitatively separate ED and MD contributions like this...these are cases where one mechanism is dominant, and so you see either a phase inversion or not. Ref. 12 is cited as justification for not considering changes in absorption between the two cases, but it is not clear where in this work that case is actually made. This is a critical point that would need to be addressed urgently and explicitly in the manuscript.

2) Regarding the justification of using the 15 mT data as a "baseline", I still find the manuscript confusing on this point, but I think maybe I understand the argument now due to the author response document. The issue is that at 15 mT they see saturation of the magnetization of the pre-existing domains, which produce a spurious effect from the demagnetization dynamics that they want to ignore. I would strongly suggest that the authors be a bit more forthcoming about this in the actual manuscript. Maybe they can show the signal at ~ 1 ps vs applied field, to show that that saturates at the same H field as the static MOKE? Currently the logic seems very circular: we are first told that "angular momentum conservation" (assuming only magnetization as a reservoir in the sample) implies that it is impossible to have H field-driven dynamics on sub- and few-picosecond time scales, and then this leads to the conclusion that there is a "delay" in the orientation...which is essentially the initial assumption back again. I think that this method of analysis in this particular section is in fact probably fine, it is more the presentation that needs fixed.

3) The larger question that follows from point (2) is whether the 20 ps "delay" is really there. In the response (but not the manuscript) the authors clarified that what they really mean is that there is a linear dependence of the emission on H only after 20 ps. This is not shown in the paper (or the comments). There is no plot of the effect vs. field at time delays of < 20 ps, and from the sets shown I don't see how one can exclude completely a small linear dependence...the data is in fact rather noisy.

I agree that there is definitely a strong increase in the field dependence after about 20 ps, but in

my view the authors are far too aggressive in the abstract and conclusions about this issue...they cannot claim there is "no" effect until 20 ps, they can only claim that it is below their detection threshold...which should be quantified if possible.

It would also be good to know how exactly the number of 20 ps for the "delay" is determined in a quantitative sense.

4) I appreciate the addition of the reference to the earlier work on single pump THz and agree that this is sufficient for this paper as justification of the claim that single pump THz emission may be from pre-existing FM domains.

5 & 9) The sweeping statements in the manuscript and the response about the differences between the current measurements and "any single pump stroboscopic measurement" are exaggerated. I would point out that exactly the effects that the authors claim are a problem for "single pump" measurements are the effects that necessitated the procedure I discussed above in point 2 (i.e. subtracting the 15 mT data). Thus the claim that "double pump" measurements are immune is inaccurate. I also note that the authors in the revised manuscript still do not give a plausible explanation for how residual FM nuclei could explain previous tr-MOKE data. In fact the analysis of their own data actually just assumes there are no orientation dynamics on sub-picosecond time scales on theoretical grounds...which is fine to present as an argument, but it is not right to give the impression that the experiment itself shows this.

I would instead argue that the main innovation of this method is to drastically increase the effective time resolution of THz emission probes of transient states, which is itself pretty nice and worth emphasising.

6) The new toy model presented for the process is much better than the original. It is not really clear to me how exactly the model shows that fluctuations drive the orientation dynamics, since I don't see anything non-deterministic in the model. I also think that the claim that the orientation "delay" is independent of the time scale of the change in J_{eff} is incorrect...it is independent as long as this time scale of change is much less than ~ 20 ps. The authors should also give the functional form of $J_{\text{eff}}(t)$ that they use for Fig. b...this seems quite a bit faster and simpler than the dynamics measured in Ref. 35 (which shows strain dynamics up to 50 ps). In addition, it would be interesting to show how slow $J_{\text{eff}}(t)$ has to be to affect the magnetization dynamics for this model.

Reviewer #2:

Remarks to the Author:

REVIEWER COMMENTS

Reviewer #1 (Remarks to the Author):

In this work the authors present a mostly experimental study of the laser-driven AFM-FM transition in FeRh, a system that has been discussed extensively in the ultrafast magnetism community. The authors apply a relatively recently developed experimental technique, what they call “double pump” THz emission. This is discussed and the data analysed in the context of a phenomenological model of the phase transition. They make several main claims in the conclusions: (1) That there is a “latency” period of 20 +/- 5 ps where the nucleation centres of new ferromagnetic domains are not susceptible to an external field, and (2) this latency time is a fundamental property of all first order magnetic phase transitions.

The general scientific questions addressed in this work are relevant, and indeed this method may yield useful insights. The authors I think overstate the controversy on this material; although early tr-MOKE measurements (ref. 4 and 5) did indeed make the surprising claim of an “ultrafast” FM phase, the subsequent work has showed that this was an artefact of the measurement technique (cf. refs. 7 and 8, the latter of which actually redoes the tr-MOKE and shows the lack of an ultrafast component). Nonetheless, it is true that the dynamics of the magnetic phase transition are not well understood and that FeRh is an excellent model system to study this.

This said, I am not convinced that the authors adequately support their main claims, and indeed their claims appear to be contradicted by existing literature and so I am not prepared to recommend this for publication in Nature Communications. In addition, there are several major apparent inconsistencies that prevent me at present from recommending this paper for publication in general. Below I list my concerns.

0) We thank the Referee for the time invested in reviewing our manuscript. The feedback helped us to significantly improve our manuscript. However, we respectfully disagree with the Referee in his statement about contradictions between our claims and existing literature. We would like to address these points below.

1) One major issue that I have is the apparent claim from the authors that performing measurements with the pump coming through the backside of the sample (through the substrate) is equivalent to performing the measurements upon spatial inversion. This is just not true, and I don't see any reasonable argument for claiming that it is. Spatial inversion would require inverting everything except the applied H field. I'm not sure why the authors didn't just use an inverted applied H field to try to separate magnetic dipole origins from electric dipole origins. The method that the authors use instead is not correct and will give misleading results, since there are pretty easy to see differences between pumping through the Pt layer vs the substrate due to the dept dependence of

the laser absorption. Thus I doubt that much of the discussion around Fig 4 is very meaningful.

1) We apologize for not being clear with our descriptions in the manuscript. Our goal is indeed to separate the THz emission of magnetic dipole and electric dipole sources, which rely on different physical mechanisms. While the former originates directly from the laser-induced ultrafast decrease of the magnetization of ferromagnetic FeRh (*i.e.* demagnetization), the latter is an indirect result of the demagnetization which, in turn, leads to spin injection into the Pt-layer and subsequent conversion of this spin-current into charge current due to the spin Hall effect ([T. Kampfrath *et al.*, *Nature Nanotech* **8**, 256–260 (2013)] and [O Gueckstock *et al.*, *Adv. Mater.* **33**, 2006281 (2021)]). This charge current is responsible for the THz emission of electric dipole origin. As both sources originate from demagnetization, however, inverting the direction of the magnetic field will invert the sign of them both. We would like to emphasize that the method applied to our work is widely used in the analysis of THz spectroscopy ([W. Zhang *et al.*, *Nature Commun* **11**, 4247 (2020)] and [T. Huisman *et al.*, *Nature Nanotech* **11**, 455–458 (2016)] and [T. Seifert *et al.*, *Nature Photon* **10**, 483–488 (2016)] and [E. Papaioannou *et al.*, *Nanophotonics*, **10** (4), pp. 1243-1257 (2020)] and [Y. Wu *et al.*, *Adv. Mater.* **29**, 1603031 (2017)] and [Qi Zhang *et al.*, *Phys. Rev. Applied* **12**, 054027 (2019)]). The discussion is therefore anything but meaningless.

2) The authors claim that “residual nuclei” of the FM phase are insusceptible to an applied field < 1 T (see lines 201-202 and 206-207), but I don’t see any support for this statement in ref. 25, which is not even about FeRh. Thus I am uncertain about the validity of the analysis procedure which is based on this claim.

2) We apologize for this misunderstanding. In fact, the sentence was meant to make a very general statement which is valid to all magnetic metals including FeRh. In particular, we would like to state that changing the orientation of a single electron spin must obey the fundamental law of conservation of angular momentum $\frac{dS}{dt} = -\gamma \mathbf{S} \times \mu_0 \mathbf{H}$, where $\gamma = 28 \text{ GHz/T}$ is the gyromagnetic ratio. Hence the characteristic strength of magnetic fields involved in the process of sub-picosecond demagnetization is of the order of 100 T (*i.e.* the characteristic fields of the exchange interaction) and magnetic fields of 1 T are simply unable to affect sub-picosecond magnetization dynamics.

3) The claim that there is no ferromagnetic signal at all for the first 20 ps is not well supported. First, as I discussed in point 2 the procedure of subtracting the 15 mT data is a bit questionable. Second, the authors do not show the data without this subtraction in a clear plot for the time period in question, which makes it difficult to assess if indeed the data have sufficient signal-to-noise to really see if anything unusual happens at 20 ps. All data have limited signal-to-noise, and it is therefore not surprising that for a slow onset effect like this that there is a period of time when it is hard to distinguish a signal from noise. That is not the same thing as a “latency” time. In fact, as I mention below (point 8), in the XMCD data of ref. 7 there is a small but measurable signal in this time

frame, which would seem to argue against the authors' main claim. Also, the authors do not give enough information about the constraints used for the linear fits in Fig 5b.

3) We respectfully disagree with the Referee and are afraid that the Referee misunderstood us. Our statement of “no ferromagnetic signal” was based on the fact, that ferromagnetic nuclei in an external magnetic field must satisfy the fundamental law of conservation of angular momentum $\frac{d\mathbf{M}}{dt} = -\gamma\mathbf{M} \times \mu_0\mathbf{H}$, *i.e.* the rate at which the magnetization changes depends linearly on the applied magnetic field strength. Figure 5 shows that such a dependence is seen only after 20 ps. As Ref. [I. Radu *et al.*, *Phys. Rev. B* **81** 104415 (2010)] does not compare measurements at different magnetic fields in the range 0-20 ps, it simply cannot contradict our finding.

The linear fit shown in Fig. 5(b) is fitted to the data points measured between 20-40 ps. We have added an additional line to clarify this constraint of the fit.

4) The authors claim that the double pump measurements show that the small signal observed in the “single pump” THz emission is from residual FM domains that are metastable at low temperatures, but I do not see any support for this. In fact, after line 93 I do not see any mention of this claim again at all.

4) We apologize for not being more clear. The statement has been also published earlier [R. Medapalli *et al.*, *Applied Physics Letters* **117**, 142406 (2020)] and we therefore thought it acquired less attention in this paper. This conclusion can be drawn from single pump THz emission experiments, where the shape and polarity of the signal is the same below and above the phase transition. Since above the phase transition, the only possible effect of femtosecond laser excitation on magnetism of a metal is sub-picosecond demagnetization, one has to accept that also below the temperature of the phase transition the signal must be caused by ultrafast demagnetization as well.

We clarified this statement and added the reference to the previous work.

5) The overall claim that earlier tr-MOKE measurements are seeing dynamics of the residual ferromagnetic nuclei (lines 269-271) is not well supported. Simply not seeing an effect in this experiment is not proof that there is nothing to see. Also, the authors do not seem to give a plausible explanation of how these residual FM nuclei could result in the tr-MOKE signals observed.

5) Thank you for pointing this out, so we can be more clear in our claims. In fact, one of the goals of our manuscript is to convince readers that the only possible process that can be active in FeRh on the sub-picosecond time scale is ultrafast demagnetization of ferromagnetic nuclei. The statement is not limited to tr-MOKE measurements only and valid for all techniques sensitive to the net magnetization.

Note that any single-pump stroboscopic measurement (not only tr-MOKE) cannot distinguish whether the signal comes from pre-existing nuclei or laser-induced nuclei.

This problem can be avoided by using a double pump experiment to separate the signal from pre-existing and newly formed nuclei.

We added these additional arguments in our revised manuscript to strengthen our claims. See lines 112-118

6) The phenomenological model is not sufficiently well explained and justified. Why do the authors assume that the lattice expands on the same time scale as the lattice temperature increase, when they themselves pointed out that this expansion happens on a time scale given by the film thickness divided by the speed of sound? I also don't see how it is reasonable to propose a model that appears to assume a homogeneous magnetic state, when for a first order transition it is clear that nucleation plays a critical role.

6) We understand the concern of the referee and apologize for the confusion. We do not claim that we have built a complete model of FeRh, but in fact we simply show that the kinetics of a system comprising essentially two macroscopic spins, where the exchange interaction is suddenly changed, can successfully reproduce the observed latency. It strongly suggests that this latency must be a fundamental property of first-order magnetic phase transitions.

We agree with the referee that it may be confusing to link the lattice expansion to both lattice temperature and speed of sound. Therefore, in the revised version we have simplified the model. Taking the time-dependency of J_{eff} as a free parameter, we focus on two regimes: an instantaneous change of J_{eff} and a change in sync with the lattice expansion. In both cases the model shows a latency, further supporting that the latency is a fundamental property of the phase transition.

7) On line 104, the authors use ref 7 as an example of previous work that also shows that the double pump change in emission field changes on a time scale of 300 ps. However, this reference does not report THz emission at all, it is a X-ray MOKE study. I get that the magnetic dynamics inferred from the measurement are similar, but the way the reference is worded the reader is led to believe that this is a THz emission study.

7) We thank the Referee for pointing this out and have corrected this in the revised version.

8) The authors misstate the overall conclusions from Ref. 7. This work does not show that there are no changes in the XMCD signal of FeRh up to 10 ps. This number of 10 ps was the time resolution of some of the experimental data, but the data in Fig. 3 do indeed show that in the overlap region between the pump and probe for the 10 ps duration probe pulse there is a measurable XMCD change.

8) We thank the Referee for pointing this out and have revised our statements.

9) In the introduction the authors discuss the limitations of stroboscopic measurements, which is reasonable in principle but they also seem to imply that their measurements are somehow exempt from the problems presented. This is misleading, in my opinion. Despite the authors' claims that the "double pump" method is fundamentally different, in fact it is really just a different type of stroboscopic pump-probe measurement where the second "pump" is really a probe. I also don't see how the authors' experiments cannot be influenced by hysteresis effects.

9) We apologize for this confusion. We strongly disagree with the Referee that our method is in fact single-pump, double probe method. The fundamental difference with a conventional pump-probe experiment is that here we use not two, but three pulses which define the measured signal where the two pump pulses have different repetition rates. More particularly, the first pump generates nuclei that evolve in the following time domain towards a ferromagnetic state, the second pump pulse demagnetizes these ferromagnetic nuclei and causes their THz emission which is read out by the third (probe) pulse using the principle of electro-optical sampling. In this way, our detection technique is focused on the nuclei generated by the first pulse and can follow their evolution over time.

The important difference is that the single pump technique will collect signals from both the demagnetization of the pre-existing ferromagnetic domains as well as the emerging magnetization of the nuclei generated by this pump. Our technique allows one to distinguish signals from the pre-existing and newly generated ferromagnetic nuclei. In this respect, co-existing ferromagnetic and antiferromagnetic phases and, consequently, temperature hysteresis do not hamper our measurements.

Minor comments, which can be optionally addressed:

10) The authors should give at least a brief explanation of the parameters of the single pump and double pump measurements in the main text, before the discussion of results. It is not clear for many of the data important details like the orientation and strength of the applied H field, or the fluences of the pulses. I know some of this is given in the Methods section, but it should be possible to follow the paper without flipping back and forth.

10) We thank the Referee for the suggestions. We have made the proper changes.

11) Overall I would suggest that the authors minimize the interpretation of the data when first discussing the results. The argumentative style of writing when new data are presented is distracting and makes it difficult to assess what is directly from the measurement and what is an interpretive argument from the authors.

11) We thank the Referee for this suggestion and made additional changes.

Reviewer #2 (Remarks to the Author):

The authors of this manuscript provided a detailed study of the AFM-to-FM phase transition of FeRh utilizing a double-pump THz emission setup. By variation of the sample temperature, external magnetic field, and excitation geometry they found a significant dependence of the THz emission strength on the presence of the FM phase in FeRh. The authors verified their findings by comparing to FeRh sample with Pt and Au cap layer. The authors found two regimes of the photo-driven phase transition: 1. a 10-20 ps delay of the emergence of the FM phase after photoexcitation, which might be a “fundamental property” of the phase transition and 2. a magnetic field dependence of the growth velocity of FM domains lasting a few 100 ps. The experimental findings were partially resembled by a macro-spin model accounting for magneto-structural interactions.

Since I have several technical/methodical as well as physical concerns with the current form of the manuscript, I cannot recommend publication without major changes and clarifications.

0) We thank the Referee for critically examining our manuscript. The feedback has helped us to improve our manuscript greatly. We will address the Referees comments point-by-point below.

General remarks:

1. I am a bit confused by the term nuclei which is used by the authors to describe the nucleation of FM domains. I would suggest either explaining the current meaning of this term in their context or changing the wording to FM domain nucleation instead.

1) We thank the Referee for this valuable suggestion. We have added additional clarification on this term at the introduction of our manuscript, as proposed by the Referee.

2. In my view there can be two pathways from the AFM towards the FM phase: either “directly” by realigning the AFM spins to local FM order which then grows to larger domains. I guess this is what happens in the temperature-driven phase transition. So in essence there is always local magnetic order present. But as a second route, one could completely destroy any magnetic order starting from the AFM phase using a fs-laser pulse and then recover from above the Curie temperature of the FM phase. Do the authors have any arguments for or against one or the other scenario? Please remind yourself, that the excitation in the sample will not be homogeneous, neither in- or out-of-plane because of the Gaussian beam profile and exponential absorption profile.

2) Thank you for this insightful question. Our interpretation is that the laser pulse induces a strain on the lattice resulting in a local change in sign of the effective exchange interaction. The antiferromagnetically ordered spins then realign locally to a

ferromagnetic order which then grows with time. In the proposed model, magnetic order is never completely destroyed. However, strictly speaking the arguments for one or the other scenario mentioned by the Referee can only be found by detecting the ultrafast dynamics of the antiferromagnetic Neel vector. We are not aware of such measurements in FeRh.

But also the AFM order will be quenched by the laser, as there is no selective excitation of the lattice. So exclusively linking the complete dynamics of the phase transition to a strain effect seems not appropriate to me.

3. The authors state that the THz double pump emission spectroscopy has the advantages of a more defined relaxation of the sample state, compared to conventional pump-probe approaches. I actually do not see that! The authors also do a typical pump-and-probe experiment at 1kHz rep. rate and the sample must relax back to an initial and defined state. The only difference is that their probe is generated by another pump. In my opinion, this only leads to a double-excited state which could result in a more defined final state of the dynamics, in case of reaching some kind of saturation of the dynamics, e.g. the Curie temperature of Fe. But the same could be also achieved by strong enough pump pulses in e.g. MOKE setups. On the other hand, I agree that the THz emission spectroscopy is a rather sensitive method to probe FM order, which, however, can also have non-magnetic contributions. So I don't see the point why this experiment could not be done with MOKE or Faraday setups? Could the authors even carry out conventional MOKE measurements with their setup?

3) We apologize for not clarifying this confusion. The fundamental difference with a conventional pump-probe experiment is that here we use not two, but three laser pulses where the two pump pulses differ in repetition frequency. In our case, the first pump generates nuclei of ferromagnetic domains that evolve in the following time domain towards a ferromagnetic state, the second pump pulse demagnetizes these ferromagnetic nuclei which causes THz emission that is read out by the third (probe) pulse using the principle of electro-optical sampling. In this way, our detection technique yields a study of the ferromagnetic nuclei generated by the first pulse and follows their evolution in time.

I think referee 1 and me are both aware of how EOS works, but essentially the third sampling pulse is "just" part of your detector/spectrometer. So we are left with the situation that a first pulse excites some dynamics and the second pulse generates a detectable signal.

The important difference is that a single pump-probe technique will collect signals from both the ferromagnetic domains, which already existed before the arrival of the pump, and the ferromagnetic nuclei generated by this pump. In contrast, our technique allows to distinguish signals from the pre-existing and newly generated ferromagnetic nuclei.

This cannot be achieved by any conventional MOKE pump-probe setups without adding an additional pump.

Any magnetically sensitive probe, e.g. MOKE, will be sensitive to residual FM domains simply as they exist already before any excitation. The authors also state, that on sub-ps time scales only demagnetization should occur in the residual FM domains. Then the dynamics form the residual and the generated FM domains will have different signs, indicating quenching and generation of FM order. Having said that, I am still not convinced by the advantages of the THz double pump technique.

“Hence, the next pump pulse can trigger two types of dynamics: from the antiferromagnetic or from the metastable ferromagnetic state. In principle, the signals of these two types cannot be distinguished by a single-pump experiment.”

In a shot to shot evaluation of any pump-probe experiment, one would observe a large difference for the magnetic signal before time zero, which could be handled with proper data analysis.

So as the authors and myself stated, any kind of excitation that resets the sample to a defined state (second very strong pump pulse) could be used here. Obviously, such second pump pulse is intrinsically present in the THz double pump technique.

4. The authors give an overview of the controversial previous results on the phase transition of FeRh, which also triggered their study. However, I miss a thorough discussion of the discrepancies between the different studies and the results of this study. I can imagine that different techniques probe different sample volumes and that the sample geometry might also have an influence on the dynamics.

4) Thank you. We have added further discussion on the different studies.

It would be very helpful to somehow indicate these changes.

5. In the last experimental part, the authors subtract low-field data (15 mT) in order to get rid of residual FM and non-magnetic contributions. Shouldn't it be possible to use zero-field data here, as both contributions are discussed to be not influenced by external magnetic fields?

Or alternatively, why didn't the authors toggle the external field and use half of the difference signal as extracted transient magnetic signal? Again the two former contributions should not depend on low magnetic fields below 1 T.

5) Toggling the external magnetic field and using half of the difference signal is indeed an alternative way to extract the transient magnetic signal. However, this method would also include the signal coming from the ultrafast demagnetization of pre-existing ferromagnetic domains since this is also magnetic in origin. Therefore, in order to remove any magnetic signal coming from the pre-existing domains, we subtract the signal at the magnetic field strength of 15 mT since this saturates all pre-existing domains along the magnetic field. Another way to remove the pre-existing domains is to work at lower temperatures which gets rid of pre-existing ferromagnetic domains.

6. The authors compared front- and back-side pumping of the samples and noted that reflection and absorption effects can be neglected. Is there any measurement or estimation for this assumption? In the case of Pt-side pumping, I expect a significant deposition of energy in the Pt layer compared to back-side pumping. To that end, the magnetization dynamics should have very different amplitudes and the THz emission from a hot or cold Pt layer should be also very different as its conductivity changes.

6) We have reported the THz emission on the front- and back-side pumping in the supplementary material of ref. [Medapalli *et al.*, *Appl. Phys. Lett.*, **117** (14), 142406 (2020)]. The THz emission signal at 300K showed no significant difference in the THz signal strength between the front- and back-side pumping. We have added these arguments in the revised manuscript.

7. I am not very convinced by the theoretical model, which the authors present. They state that it results in a delay of about 10 ps even for an instantaneous change of the exchange interactions. The change of the exchange interaction is linked to the expansion of the lattice, which the authors correctly link to the sound velocity of the FeRh layer (L279). Later in L320 they discuss a sub-picosecond timescale of the change of the exchange interaction due to an ultrafast heat-induced lattice expansion. But there is no sub-ps lattice expansion in a 40 nm thick film! There can only be quasi-instantaneous STRESS but not STRAIN! Based on this fact the change of the exchange interaction will take also several ps and the authors cannot distinguish this effect from any “fundamental” delay in the phase transition. In order to do so, the authors must study the FeRh thickness or probing-depth dependence of this delay or have an additional structural probe at hand.

To this end, I am convinced that the proposed model is not correct or simply assumes way too few atomic layers (which of course need much less time to expand than a 40 nm thick film).

Accordingly, there might be no fundamental delay of the phase transition, but it is simply limited by the structural phase transition and further delayed by the growth of the FM domains.

7) We apologize for the confusion. The “sub-picosecond timescale” of the lattice expansion is a misprint, it should be picoseconds and we thank the referee for pointing that out.

We do not claim that we have built a complete model of FeRh, we simply show that a two spin system, where the exchange interaction is suddenly changed, can successfully reproduce the observed behaviour of the latency. To make this point more clear, we have simplified the model where we will show the magnetization dynamics for two situations: (i) the exchange interaction changes instantaneously and (ii) the exchange interaction changes at the timescale of the speed of sound. This is done to distinguish the fundamental delay in the phase transition from the change in the exchange interaction due to lattice expansion. We have estimated the characteristic time of the structural changes based on the speed of sound reported in [D. W. Cooke *et al.*, *Phys.*

Rev. Lett. **109**, 255901 (2012)]. This characteristic time (~10 ps) also corresponds to the lattice dynamics reported in literature. In both cases a latency is observed after the change of the exchange interaction. This strongly suggests that this latency must be a fundamental property of a first-order magnetic phase transition.

I would still insist to compare two samples of different film thickness in order to validate the structural influence on the observed delay. As the authors discuss a “fundamental” latency during a 1st order phase transition, the lattice contribution must be determined more accurately. Moreover, the results of the proposed model, as shown in Fig. 9, include the delay of the generation of FM order but already the shape of the $M(t)$ curves and their field dependence strongly disagree with the experimental data and the discussion in the manuscript. The authors state that the slope of the THz signal rise (which is proportional to the FM magnetization), see Figure 5, must be proportional to the external magnetic field. But already this is not included in their model. Instead the model results in a shift of the onset and one could also argue about the linearity of the $M(t)$ curve, as discussed for the experimental data. I am very much aware of the fact, that the authors cannot build a “complete” model, but in my opinion, such large discrepancies between experiment and model do not support their findings too much.

8. I was also wondering if the authors can give any estimate on the actual delay, between the max. of the emitted THz pulse and the actual pump-pulse. I think this delay is already reasonable and at least in the range of 10 % of the proposed fundamental delay – so it must be discussed here as well.

8) In our double pump THz emission experiment we detect the THz emission triggered by pump 2 while the change of the THz electric field amplitude is induced by pump 1 (the actual pump pulse). In our setup the time-overlap between the THz peak amplitude triggered by pump 2 and the probe pulse is fixed and we scan by varying the delay of pump 1. Therefore, the response of the change in the THz peak amplitude is the same as the temporal resolution of our setup which is about 50 fs.

9. What is the FM saturation field strength to obtain a single-domain state above the phase transition temperature? Is there an estimate about the domain sizes below the saturation field?

9) The ferromagnetic saturation is 15 mT. We have measured the magnetic hysteresis curve of FeRh in the ferromagnetic phase at 400 K and added to in the supplementary file. We cannot make any estimation of the domain sizes in our sample, but according to Ref. [C Baldasseroni *et al.*, *J. Phys.: Condens. Matter* **27**, 256001 (2015)] the domain size in similar samples is of the order of 1 μm .

10. Why would the fundamental delay of the phase transition be so different in the Pt vs. the Au sample? I think there is a significant difference between the two experimentally obtained values? The presented theory does not take the cap layer into account at all.

10) To understand the origin of the difference in the latency for FeRh/Au and FeRh/Pt, one has to be able to experimentally distinguish exchange-driven spin dynamics from the dynamics of the exchange interaction. However, this is not yet possible and the difference remains unclear to us. Moreover, our model only takes into account the spin systems and lattice expansion of FeRh. We did not include any capping layers.

So what do we learn from the FeRh/Au results? The difference in the latency for Pt and Au capping is significant and there is no discussion of this at all – it is just noted, that the latency is of “same order of magnitude”. The author should discuss this issue in the manuscript.

Specific comments:

L81: replace “dramatic” with a more meaningful/quantitative adjective, like “100 times” or “significant”

We revised the word to “significant”.

L95-96: Both pump pulses initiate magnetization dynamics in a broader sense. The authors might differentiate between AFM and FM magnetization dynamics or need to be more specific.

We revised it to “If the first pump generates ferromagnetic nuclei, then the second pump initiates the ultrafast demagnetization of these nuclei, which results in THz emission.”

L118: again the authors should carefully distinguish between FM and AFM “magnetization” as also the AFM phase has a sub-lattice magnetization.

We revised the paragraph and clarified that the first pump launches a process opposite to the ultrafast demagnetization of the ferromagnetic phase *i.e.* the growth of the ferromagnetic phase in FeRh.

L132: please describe the “directions” of the magnetic field here.

We added additional explanation that an in-plane magnetic field was applied.

Figure 3: The electric field strength in (b) at 120 mT is smaller than in (a) for 105 mT. This is not consistent and must be explained.

The two datasets of Fig. 3(a) and 3(b) were taken weeks apart. Small day-to-day fluctuations in the alignment in the setup resulted in a lower ΔE peak at the highest magnetic field strength.

The normalization of the field strength in Fig. 3 covers the issue with the day-to-day fluctuations, but it also misses the H-field dependence of the THz field amplitude as this is related to better alignment of the FM domains along the applied H-field. I would prefer the former version of the figure with a short statement about amplitudes in the methods section.

Figure 4: again the field strength in (a) 300 K is different than in Figure 3.

The same as with Fig 3, fluctuations in the alignment resulted in a lowered field strength. Some dataset were taken weeks to months apart.

Line 179-80: $dE(t=x)$ will be zero at the point where dE changes sign. Hence Eq. (1) will be multiplied by zero!?

We apologize for the confusion, $\Delta E(t=x)$ is the MgO-side pumped ΔE signal. This is not zero. We take the value of the MgO-side pumped ΔE signal at the timestamp $t=x$ where the Pt-side pumped THz signal is zero. We have clarified our equation in our revised manuscript.

L215 & Figure 5: The authors state that their fit function results in a large mismatch with the experimental data. This is obvious, as they do not include the delay of the exponential rise. Please fit the data with a double exponential fit function that includes a delay. Otherwise the inset in Fig. 5 (a) is also rather meaningless. Please also scale the inset to maximize the view on the presented data.

In (b) one should not fit a linear function in order to determine the delay of the onset. Please use again a step- or error-function in combination with a linear function instead.

We want to emphasize that the fit was first done with the understanding that there were only two regimes of dynamics, the rise and relaxation. The mechanism of the latency is still not understood well enough to simply fit it with some modified function. Modifying the function without a good physical reason would also change the other extracted fit parameters that will put doubt in the overall explanation of the fit.

I disagree here! Adding a delay to the fit function is obvious and is a first order approximation for any physical mechanism which causes this latency. This is a clear experimental finding and the reasoning is done in a second step.

I think it is not appropriate to fit data with an obviously wrong model just to show that it is wrong, when a rather simple analytical model is at hand.

At the same time, including the delay into the fits, would give a more objective value of the latency time. As seen from Fig. 5b, it seems that the determined latency of 20ps is not a result of a fit but read by eye and is kept constant for different magnetic fields.

I think that this value must be determined in a more objective manner. The same applies for the delay of the FeRh/Au sample as shown in Fig. 7.

We removed the overall fit of the data in Fig. 5(a) and instead added this as a supplementary figure.

Figure 7: please bin the data and add error bars instead of cluttering the complete plot with data points.

Again the fits in (a) and (b) do not look very convincing to me.

We have binned the data in Fig. 7 and removed the fits in Fig. 7(a).

How about errorbars?

Figure 9: Can one neglect heat diffusion out of the phonoic system of FeRh into the Pt cap and MgO substrate layer? The FM saturation is reached already after 30 ps in the simulation, while the experiment needs about 300 ps, can you discuss the discrepancy? Why does the FM saturation moment not depend on the external magnetic field as in the experiment. Moreover it is only 0.8 μB and not the full 3.8 μB of Fe. I think the 3.8 μB for Fe is also a rather large value.

Please also shift the time zero from 3 ps to 0 ps in accordance to the experiment.

Accordingly, the onset at 10 ps is only 7 ps after the first pump pulse in the simulation and not approx. 10 ps!

I would also appreciate the usage of experimental SI units even for the simulations.

We focused in our model on the qualitative description of the transition to the ferromagnetic state, in particular the latency behavior. Therefore, we did not include any heat diffusion processes. Moreover, we consider a homogeneous heating across the whole sample, which is justified by the fact that the film is only 40 nm. This results in the ferromagnetic saturation and discrepancy in the timescale seen in the simulation. In our revised version, we have simplified our model and normalized the saturation moment of Fe for clarity.

Normalizing the magnetization does not resolve the discrepancy between your model and the general theoretical expectations regarding the magnetic moment. I am still missing a discussion about the discrepancy of FM saturation times between the model (30ps) and the experiment (300ps). I guess there a good reasons for that, but it needs to be discussed in the manuscript.

L 321: heath = heat

This misspelling has been corrected

L395: the second pump pulse should be much smaller than the first one, in order to probe a homogeneously excited region with no in-plane gradients.

We chose to keep the beamspot equally large to maximize the signal-to-noise ratio. Moreover, a smaller spot of second pump pulse did not affect the overall dynamics.

Here I am a bit puzzled, as I would strongly expect a change of the FM response for different pump2 spot sizes as this is essentially a variation in the pump fluence. Did the authors measure any fluence dependence at all?

L402-03: this assumes that the delay, phase, and spectrum of the THz pulses do not change throughout the delay scan. Have the authors checked this by carrying out full EOS spectra at different pump1-pump2 delays?

We have provided below the THz electric field trace at two time-delays of the two pumps. The right graph shows that the THz waveform does not change throughout the delay scan.

Referee #1

The referee's comments are in black.
Our responses to the referee are in blue.

In their revision, the authors have partly addressed and responded to some of the points raised earlier. While some issues are clearer, there are still several major outstanding issues with the current version that preclude me from recommending it. I discuss the points using the same numbering as in my original report.

1) A significant part of the argumentation of the paper relies upon the idea that it is possible to separate magnetic dipole and electric dipole contributions of the THz emission by performing two measurements, one where the sample is pumped from the Pt side and the other where it is pumped from the MgO side. After looking through the references provided by the authors that claim this is a “standard practice,” I suspect that one issue may be an imprecise description of the method. In each of those references, the issue of whether THz emission is predominantly from demagnetization itself or from the spin-hall effect is answered by performing measurements where the sample is rotated by 180 degrees about the applied in-plane H field. Is that what the authors actually did here? It is not clear. If I read the paper literally, the procedure is that the **only** thing that changes is to realign the pump so that it pumps the sample from the MgO side, but the sample itself and the detection are unchanged. This is clearly not adequate for differentiating magnetic dipole (MD) from electric dipole (ED) contributions, since what really needs to change is the sample orientation relative to the detector. The authors would at a minimum need to clarify this issue. **Assuming** that what is really happening here is that the sample is rotated by 180 degrees (flipped), it is still an open question if this really makes it possible to quantitatively differentiate MD and ED contributions. This makes a number of other assumptions that are not clearly stated and also not clearly justified. One such assumption is that the two pump pulses excite the sample with the same spatial profile for both geometries. Can the authors justify this? I suspect it might be difficult, since ~20 nm is the approximate attenuation length of the pump in FeRh. In reviewing the papers cited by the authors as previous applications of their method, I did not find any that tried to quantitatively separate ED and MD contributions like this...these are cases where one mechanism is dominant, and so you see either a phase inversion or not. Ref. 12 is cited as justification for not considering changes in absorption between the two cases, but it is not clear where in this work that case is actually made. This is a critical point that would need to be addressed urgently and explicitly in the manuscript.

- **Response:** We are very grateful to the referee for the very careful and critical look at our paper. We apologize for not making the point about the sample orientation more clear. We indeed rotated the sample by 180 degrees about the applied H-field and this sample orientation change is relative to the detector. We have added an additional sentence in lines [172 - 175] to clarify this:

“In our experiments the pump was incident either from the Pt or MgO face of the sample and the detected THz signal was emitted from the MgO or Pt face, respectively. We switched between these two configurations by rotating the sample by 180 degrees around the applied in-plane magnetic field.”

We agree with the referee that the employed procedure to differentiate the MD and the ED signal is not free of uncertainties, because the experimental geometries of pumping from the Pt and MgO faces are not equivalent. Next to the spatial profile of the deposited energy in FeRh film mentioned by the referee, in these two configurations visible pump light is reflected differently from Pt and MgO sample faces. Moreover, also the emitted THz waves follow different paths from the source to the detector.

In our experiments, the non-equivalent spatial profiles of the two geometries can result in a difference of the amplitudes and temporal profile of the signals. In order to account for the non-equivalence of the geometries, resulting in a difference in amplitudes of the THz signal, we first identified the time points, when the double pump THz signal changes sign (see, for instance, the blue line in the top panel of Fig. 4(a)). At these time points, the MD and the ED signal cancel each other out due to destructive interference. In the procedure of quantitative estimates of the amplitudes of the ED and the MD THz emissions, we used these points as the reference and renormalized the data obtained in the experiments with pumping from the MgO-side, using Eq. 1.

$$\Delta E_{(t)}^{\text{MgO}'} = \text{sgn}(T_{\text{PT}} - T) \times \frac{\Delta E_{(t)}^{\text{MgO}}}{\text{Max}(|\Delta E_{(t)}^{\text{MgO}}|)} \times \Delta E_{(t=x)}^{\text{MgO}}. \quad (1)$$

Figure 4 the left panel at 300 K, the blue data points switching sign at roughly 0.4 ns. The same can be seen for the data points at 360 K.

To account for the temporal profiles, THz traces measured for the very same sample for pumping from the Pt and MgO sides, respectively, are given in supplementary figure 2 in Ref. [R. Medapalli et al. Appl. Phys. Lett. **117**, 142406 (2020)]. It shows in fact that their temporal profiles are equivalent.

Supplementary Figure 2 from Ref. [R. Medapalli et al. *Appl. Phys. Lett.* **117**, 142406 (2020)]

Finally, and most importantly, the main goal of the differentiation of the MD and the ED signals is the motivated choice of the experimental geometry in which these two sources interfere constructively and thus result in a superior signal-to-noise ratio. Other unaccounted uncertainties in the quantitative differentiation, if any, hardly affect the main conclusions of the paper.

Following the request of the referee, we have commented on this issue and added the discussion about the effect of non-equivalent geometries in lines [202 - 209].

“When differentiating the magnetic- and electric-dipole sources non-equivalent spatial profiles at the MgO- and Pt-side can result in a difference in the amplitude and temporal profile of the THz signals. To account for the difference in THz signal amplitudes, we first identified the timestamp, when the ΔE signal changes sign (see the blue line in the top panel of Fig. 4(a)). At this time, the magnetic- and electric-dipole signal cancel each other out due to destructive interference. We use these timestamps as a reference and renormalized the THz signals obtained in the experiments with pumping from the MgO-side with the following equation:”

And lines [217 - 221].

“For the temporal differences due to possible non-equivalent spatial profiles, THz waveforms pumped from the Pt- and MgO-sides, respectively, were measured for a similar sample in Ref. [12, N. Supplementary figure 2]. The near identical waveforms show that the temporal differences are negligible. This also means that absorption

and reflection effects in the MgO-substrate and Pt-film did not influence the THz waveforms.”

2) Regarding the justification of using the 15 mT data as a “baseline”, I still find the manuscript confusing on this point, but I think maybe I understand the argument now due to the author response document. The issue is that at 15 mT they see saturation of the magnetization of the pre-existing domains, which produce a spurious effect from the demagnetization dynamics that they want to ignore. I would strongly suggest that the authors be a bit more forthcoming about this in the actual manuscript. Maybe they can show the signal at ~ 1 ps vs applied field, to show that that saturates at the same H field as the static MOKE? Currently the logic seems very circular: we are first told that “angular momentum conservation” (assuming only magnetization as a reservoir in the sample) implies that it is impossible to have H field-driven dynamics on sub- and few-picosecond time scales, and then this leads to the conclusion that there is a “delay” in the orientation...which is essentially the initial assumption back again. I think that this method of analysis in this particular section is in fact probably fine, it is more the presentation that needs fixed.

- **Response:** We apologize for the confusion caused. We have rephrased the paragraph in lines [237 - 254] trying to make our motivation for subtracting the 15 mT data more clear.

“At the (sub-)picosecond timescale, the law of conservation of angular momentum dictates that the speed of the magnetization dynamics ($\frac{d\mathbf{M}}{dt}$) scales with the magnetic field (\mathbf{H}) as $\frac{d\mathbf{M}}{dt} = -\gamma\mathbf{M} \times \mu_0\mathbf{H}$, where $\gamma = 28$ GHz/T is the gyromagnetic ratio. Therefore, picosecond magnetization dynamics can only occur in effective magnetic fields of the order of 10 T and similarly strong applied fields are needed to affect such dynamics. At external magnetic fields much less than 10 T, the dynamics of sub-picosecond demagnetization of ferro- and ferrimagnetic metals does not depend on the magnetic field strength [25]–[27]. This should also be the case for the ferromagnetic nuclei in FeRh.

The single-pump THz emission performed on the sample, reported in Ref. [12, Fig. 4], showed a magnetic hysteresis above T_{PT} corresponding to the MOKE signal in Fig. S3 (see supplementary), where the magnetization saturates above 15 mT. Hence, if any residual ferromagnetic domains are present in the predominantly antiferromagnetic phase and a femtosecond laser pulse causes their sub-picosecond demagnetization, the latter should not be affected by fields above 15 mT. Therefore, in order to remove the part of the signal originating from sub-picosecond demagnetization of the residual ferromagnetic domains as well as signals from non-magnetic sources, we took data measured at 15 mT as a baseline and subtracted it from measurements obtained at higher field strengths.”

3) The larger question that follows from point (2) is whether the 20 ps “delay” is really there. In the response (but not the manuscript) the authors clarified that what they really mean is that there is a linear dependence of the emission on H only after 20 ps. This is not shown in the paper (or the comments). There is no plot of the effect vs. field at time delays of < 20 ps, and from the sets shown I don’t see how one can exclude completely a small linear dependence...the data is in fact rather noisy.

I agree that there is definitely a strong increase in the field dependence after about 20 ps, but in my view the authors are far too aggressive in the abstract and conclusions about this issue...they cannot claim there is “no” effect until 20 ps, they can only claim that it is below their detection threshold...which should be quantified if possible.

It would also be good to know how exactly the number of 20 ps for the “delay” is determined in a quantitative sense.

- **Response:** In response to the referee, we have changed the formulations and now extracted a latency value from our fit.

Figure 5(b) the figure shows the fit to the data at the early timescale.

In order to make our claims more quantitative, we fitted the data shown in Fig. 5(b) with a function that was inspired by the Johnson-Mehl-Avrami-Kolmogorov equation. We extracted both the growth rate and latency. The fits and the fit results are shown in Fig. 5 for FeRh/Pt and in Fig. 7 for FeRh/Au of the latest version of the manuscript. The fits allow us to deduce the latency period, which is clearly present. We have added this discussion in the lines [255 - 267] for FeRh/Pt:

“The ΔE signal in Fig. 5 was normalized (explained in supplementary(4)) such that it represents the ferromagnetic fraction. We found that it was insufficient to fit our data

with the sum of an exponential rise and decay function to extract the rise and relaxation times, respectively, as was done in Ref. [7]. The large mismatch especially below 100 ps suggests that there must be an additional regime where there is a latency in the dynamics of ΔE . We therefore fitted the ΔE signal with a function which was inspired by the Johnson-Mehl-Avrami-Kolmogorov equation (explained in supplementary(5)) [28, pp. 16–21], [29], [30]

$$f(t) = \mathcal{H}(t - t_{latency}) \left(1 - e^{-K(t-t_{latency})^2}\right). \quad (2)$$

Here $\mathcal{H}(t)$ is the Heaviside-step-function and $t_{latency}$ is the latency where before $t_{latency}$ there is no ΔE signal. After $t_{latency}$, the speed of the growth of the ΔE signal is determined by K . Figure 5(b) shows the fit based on eq. (2) with the data for three different field strengths below 100 ps. The inset of Fig. 5(a) shows that the growth rate, K , is linearly proportional to the magnetic field strength which suggests that K describes the growth of the ferromagnetic phase. The estimated latency as a function of applied field is shown in the inset of Fig. 5(b).”

And lines [294 - 301] for FeRh/Au:

“Since the FeRh/Pt and FeRh/Au interfaces must result in different anisotropies [32], [33] and magnetization dynamics also depends on magnetic anisotropy for both the antiferromagnetic and ferromagnetic phases, it would be interesting to see how the kinetics of the magnetic phase transition is affected by differences in capping layer. The latency for FeRh/Au, where the THz emission is dominated by magnetic dipole sources, is shown in the inset of Fig. 7(b). The error margins are much higher compared to FeRh/Pt due to the low signal-to-noise ratio. However, the data does suggest the presence of a latency regime nonetheless.”

We revised lines [13-15] to:

“Our results strongly suggest a latency period between the initial pump-excitation and the emission of THz radiation by ferromagnetic nuclei.”

We revised Lines [392 - 393] to

“In summary, our results show a latency between the laser-induced heating of FeRh and the emergence of a net magnetization at its antiferromagnetic to ferromagnetic phase transition.”

4) I appreciate the addition of the reference to the earlier work on single pump THz and agree that this is sufficient for this paper as justification of the claim that single pump THz emission may be from pre-existing FM domains.

5 & 9) The sweeping statements in the manuscript and the response about the differences between the current measurements and “any single pump stroboscopic measurement” are exaggerated. I would point out that exactly the effects that the authors claim is a problem for “single pump” measurements are the effects that necessitated the procedure I discussed above in point 2 (i.e. subtracting the 15 mT data). Thus the claim that “double pump” measurements are immune is inaccurate. I also note that the authors in the revised manuscript still do not give a plausible explanation for how residual FM nuclei could explain previous tr-MOKE data. In fact the analysis of their own data actually just assumes there are no orientation dynamics on sub-picosecond time scales on theoretical grounds...which is fine to present as an argument, but it is not right to give the impression that the experiment itself shows this.

I would instead argue that the main innovation of this method is to drastically increase the effective time resolution of THz emission probes of transient states, which is itself pretty nice and worth emphasizing.

- **Response:** We apologize for this apparent overstatement and corrected the claims accordingly.

In the present version of the paper we do not claim that double pump experiments are immune to artefacts originating from co-existence of several phases, but “better suited” for studying kinetics of first order phase transitions. We discuss this in lines [57 - 65].

“Although the double-pump THz emission spectroscopy is not immune to artefacts from co-existing phases, we will show that it is better suited for studying the kinetics of first order phase transitions. In FeRh, where ferromagnetic domains can co-exist in the antiferromagnetic phase, both pumps will launch either ultrafast nucleation of new ferromagnetic domains or demagnetization of residual ferromagnetic domains. By measuring the THz emission as a function of the two pump delay, we can deduce the magnetization dynamics of the ferromagnetic phase transition induced by the first pump. Moreover, this technique allows one to detect magnetization dynamics up to nanosecond timescales without sacrificing any sensitivity or temporal resolution.”

We also changed our claims that our experiments show that residual FM nuclei could explain previous tr-MOKE data. Instead we now claim that based on our experiments and theoretical grounds we propose that the previous tr-MOKE data can be explained by residual FM nuclei”. We revised this in the lines [159 – 163] and [230 - 231] and [317 - 319].

“Moreover, Fig. S2 in supplementary(2) reveals fast dynamics of the ΔE signal at the first few picoseconds, which are opposite in sign with respect to the slow dynamics.

This fast dynamics is similar to the ultrafast MOKE signal reported in [5], [6]. However, the opposite sign with respect to the slow dynamics seems to suggest that the fast dynamics originates from the demagnetization of the residual domains”

“the sub-picosecond behavior can be explained as the ultrafast magnetization dynamics of the residual ferromagnetic domains”

“On this basis, the sub-picosecond magnetization dynamics in FeRh reported earlier [5], [6] can be explained by ultrafast demagnetization of residual ferromagnetic domains.”

We also thank the referee for the suggestion to emphasize the uniqueness of our detection method and have revised the lines [9 - 13] to emphasize this:

“Employing double-pump THz emission spectroscopy has enabled us to dramatically increase the temporal detection window of THz emission probes of transient states without any loss of resolution. It allows us to study the kinetics of emergent ferromagnetism from the femtosecond up to the nanosecond timescales in FeRh/Pt bilayers.”

6) The new toy model presented for the process is much better than the original. It is not really clear to me how exactly the model shows that fluctuations drive the orientation dynamics, since I don't see anything nondeterministic in the model. I also think that the claim that the orientation “delay” is independent of the time scale of the change in J_{eff} is incorrect...it is independent as long as this time scale of change is much less than ~ 20 ps. The authors should also give the functional form of $J_{\text{eff}}(t)$ that they use for Fig. b...this seems quite a bit faster and simpler than the dynamics measured in Ref. 35 (which shows strain dynamics up to 50 ps). In addition, it would be interesting to show how slow $J_{\text{eff}}(t)$ has to be to affect the magnetization dynamics for this model.

- **Response:** Our two spin model is indeed deterministic. However, in real samples a change of the isotropic exchange interaction will drive reorientation in a random direction, defined by fluctuations in every particular part of the sample. We corrected this explanation in the main text of the paper in the lines [332 - 334] and [340 - 341].

“We propose that J can be changed by strain effects which will drive the dynamics and the fluctuations and cause the reorientation of spins in random directions.”

“To gain a qualitative understanding, we consider a phenomenological and deterministic model of two macroscopic spins with magnetization \mathbf{M}_1 and \mathbf{M}_2 , respectively”

Following the suggestion of the referee, in the supplementary discussion we added an additional figure with the exchange J_{eff} gradually varying within 30 ps and modified the text in the supplementary in lines [371 - 372]:

“If the exchange integral J_{eff} gradually changes within 30 ps [9], the latency window also extends, practically following J_{eff} .”

Figure S6 The net magnetization $M(t)$ at a fixed applied magnetic field for the cases where the exchange constant $J_{\text{eff}}(t)$ changes at the timescale of 10 and 30 ps.

Referee #2

The referee's first comments are in black.

Our first responses are in orange.

The referee's responses are in red.

Our second responses to the referee are in blue.

2. In my view there can be two pathways from the AFM towards the FM phase: either "directly" by realigning the AFM spins to local FM order which then grows to larger domains. I guess this is what happens in the temperature-driven phase transition. So in essence there is always local magnetic order present. But as a second route, one could completely destroy any magnetic order starting from the AFM phase using a fs-laser pulse and then recover from above the Curie temperature of the FM phase. Do the authors have any arguments for or against one or the other scenario? Please remind yourself, that the excitation in the sample will not be homogeneous, neither in- or out-of-plane because of the Gaussian beam profile and exponential absorption profile.

2) Thank you for this insightful question. Our interpretation is that the laser pulse induces a strain on the lattice resulting in a local change in sign of the effective exchange interaction. The antiferromagnetically ordered spins then realign locally to a ferromagnetic order which then grows with time. In the proposed model, magnetic order is never completely destroyed. However, strictly speaking the arguments for one or the other scenario mentioned by the Referee can only be found by detecting the ultrafast dynamics of the antiferromagnetic Neel vector. We are not aware of such measurements in FeRh.

But also the AFM order will be quenched by the laser, as there is no selective excitation of the lattice. So exclusively linking the complete dynamics of the phase transition to a strain effect seems not appropriate to me.

- **Response:** We thank the referee for this critical look. In response, we would like to note that the scenario which involves all three phases (AFM, FM and paramagnetic PM) seems to be highly improbable to us. With the fluences of the femtosecond laser pulses used in our experiment, even the FM order in the high-temperature phase was not quenched completely. This is shown by Radu et al. [I. Radu et al., Phys. Rev. B, 81, 104415 (2010)] where the FM state was only demagnetized by 40% at similar laser fluences. Hence we do not see any reason why such pulses will cause a complete loss of the magnetic order in the low temperature AFM phase.

Moreover, in our simple two spin simulations we do not specify the origin of the ultrafast change of the exchange integral, but only propose that, in the view of popular theories of the AFM-FM phase transition in FeRh, it can be changed due to strain effects.

We have added comments to the manuscript to clarify these points. See lines [332 - 334] and [336 - 339]:

“We propose that J can be changed by strain effects which will drive the dynamics and the fluctuations and cause the reorientation of spins in random directions.”

“We do not consider laser-induced quenching of the antiferromagnetic order to play a significant role in the laser-induced phase transition. This is because a complete quenching of the antiferromagnetic order is highly unlikely since the quenching of the ferromagnetic order is only 40% [7].”

3. The authors state that the THz double pump emission spectroscopy has the advantages of a more defined relaxation of the sample state, compared to conventional pump-probe approaches. I actually do not see that! The authors also do a typical pump and-probe experiment at 1kHz rep. rate and the sample must relax back to an initial and defined state. The only difference is that their probe is generated by another pump. In my opinion, this only leads to a double-excited state which could result in a more defined final state of the dynamics, in case of reaching some kind of saturation of the dynamics, e.g. the Curie temperature of Fe. But the same could be also achieved by strong enough pump pulses in e.g. MOKE setups. On the other hand, I agree that the THz emission spectroscopy is a rather sensitive method to probe FM order, which, however, can also have non-magnetic contributions. So I don't see the point why this experiment could not be done with MOKE or Faraday setups? Could the authors even carry out conventional MOKE measurements with their setup?

3) We apologize for not clarifying this confusion. The fundamental difference with a conventional pump-probe experiment is that here we use not two, but three laser pulses where the two pump pulses differ in repetition frequency. In our case, the first pump generates nuclei of ferromagnetic domains that evolve in the following time domain towards a ferromagnetic state, the second pump pulse demagnetizes these ferromagnetic nuclei which causes THz emission that is read out by the third (probe) pulse using the principle of electro-optical sampling. In this way, our detection technique yields a study of the ferromagnetic nuclei generated by the first pulse and follows their evolution in time.

I think referee 1 and me are both aware of how EOS works, but essentially the third sampling pulse is “just” part of your detector/spectrometer. So we are left with the situation that a first pulse excites some dynamics and the second pulse generates a detectable signal.

The important difference is that a single pump-probe technique will collect signals from both the ferromagnetic domains, which already existed before the arrival of the pump, and the ferromagnetic nuclei generated by this pump. In contrast, our technique allows to distinguish signals from the pre-existing and newly generated ferromagnetic nuclei. This cannot be achieved by any conventional MOKE pump-probe setups without adding an additional pump.

Any magnetically sensitive probe, e.g. MOKE, will be sensitive to residual FM domains simply as they exist already before any excitation. The authors also state, that on subps time scales only demagnetization should occur in the residual FM domains. Then the dynamics from the residual and the generated FM domains will have different signs, indicating quenching and generation of FM order. Having said that, I am still not convinced by the advantages of the THz double pump technique.

“Hence, the next pump pulse can trigger two types of dynamics: from the antiferromagnetic or from the metastable ferromagnetic state. In principle, the signals of these two types cannot be distinguished by a single-pump experiment.”

In a shot to shot evaluation of any pump-probe experiment, one would observe a large difference for the magnetic signal before time zero, which could be handled with proper data analysis.

So as the authors and myself stated, any kind of excitation that resets the sample to a defined state (second very strong pump pulse) could be used here. Obviously, such second pump pulse is intrinsically present in the THz double pump technique.

- **Response:** The referee is correct that the signals from demagnetized and newly emerged FM domains will have different signs and thus can be distinguished. However, the situation is far less trivial if one wants to distinguish an expansion of the pre-existing and a formation of new FM domains.

It is simply a fact that single-pump experiments performed so far, including those shown in Fig. 2(a) in our paper, do not distinguish signals from pre-existing and newly formed FM domains. Nevertheless, in order to avoid misunderstandings, we would like to change the phrasing. Instead of “In principle, the signals of these two types cannot be distinguished by a single-pump experiment.” (see lines [52 - 54]), now the manuscript states: “In principle, the signals of these two types were not distinguished in single-pump experiments on FeRh performed so far.”

See also issue (5 & 9) in our response to referee 1.

4. The authors give an overview of the controversial previous results on the phase transition of FeRh, which also triggered their study. However, I miss a thorough discussion of the discrepancies between the different studies and the results of this study. I can imagine that different techniques probe different sample volumes and that the sample geometry might also have an influence on the dynamics.

4) Thank you. We have added further discussion on the different studies.

It would be very helpful to somehow indicate these changes.

- **Response:** We made additional revisions to the discussion in lines [28 - 33]:

“Later, a similar study, but using picosecond X-ray pulses as a probe, revealed that the Fe dichroic signal, related to the ferromagnetic order, emerges at the timescale of 100 ps for both the Fe and Rh sublattices [7], [8]. A similar timescale of 60 ps for the magnetic response was found, but the creation of the initial nuclei was shown to happen at the same timescale as the lattice dynamics (30 ps) [9]. On the other hand, the electronic structure associated with the ferromagnetic phase reacts at a sub-picosecond timescale [10]”

7. I am not very convinced by the theoretical model, which the authors present. They state that it results in a delay of about 10 ps even for an instantaneous change of the exchange interactions. The change of the exchange interaction is linked to the expansion of the lattice, which the authors correctly link to the sound velocity of the FeRh layer (L279). Later in L320 they discuss a sub-picosecond timescale of the change of the exchange interaction due to an ultrafast heat-induced lattice expansion. But there is no sub-ps lattice expansion in a 40 nm thick film! There can only be quasi-instantaneous

STRESS but not STRAIN! Based on this fact the change of the exchange interaction will take also several ps and the authors cannot distinguish this effect from any “fundamental” delay in the phase transition. In order to do so, the authors must study the FeRh thickness or probing-depth dependence of this delay or have an additional structural probe at hand.

To this end, I am convinced that the proposed model is not correct or simply assumes way too few atomic layers (which of course need much less time to expand than a 40 nm thick film).

Accordingly, there might be no fundamental delay of the phase transition, but it is simply limited by the structural phase transition and further delayed by the growth of the FM domains.

7) We apologize for the confusion. The “sub-picosecond timescale” of the lattice expansion is a misprint, it should be picoseconds and we thank the referee for pointing that out.

We do not claim that we have built a complete model of FeRh, we simply show that a two spin system, where the exchange interaction is suddenly changed, can successfully reproduce the observed behavior of the latency. To make this point clearer, we have simplified the model where we will show the magnetization dynamics for two situations: (i) the exchange interaction changes instantaneously and (ii) the exchange interaction changes at the timescale of the speed of sound. This is done to distinguish the fundamental delay in the phase transition from the change in the exchange interaction due to lattice expansion. We have estimated the characteristic time of the structural changes based on the speed of sound reported in [D. W. Cooke *et al.*, *Phys. Rev. Lett.* **109**, 255901 (2012)]. This characteristic time (~10 ps) also corresponds to the lattice dynamics reported in literature. In both cases a latency is observed after the change of the exchange interaction. This strongly suggests that this latency must be a fundamental property of a first-order magnetic phase transition.

I would still insist to compare two samples of different film thickness in order to validate the structural influence on the observed delay. As the authors discuss a “fundamental” latency during a 1st order phase transition, the lattice contribution must be determined more accurately. Moreover, the results of the proposed model, as shown in Fig. 9, include the delay of the generation of FM order but already the shape of the $M(t)$ curves and their field dependence strongly disagree with the experimental data and the discussion in the manuscript. The authors state that the slope of the THz signal rise (which is proportional to the FM magnetization), see Figure 5, must be proportional to the external magnetic field. But already this is not included in their model. Instead the model results in a shift of the onset and one could also argue about the linearity of the $M(t)$ curve, as discussed for the experimental data. I am very much aware of the fact, that the authors cannot build a “complete” model, but in my opinion, such large discrepancies between experiment and model do not support their findings too much.

- **Response:** We are afraid that we cannot agree with the referee and we strongly disagree that the experiments proposed by the referee, although being very interesting, will fit into the present manuscript. By changing the thickness, we will also affect the effective magnetic anisotropy in both the AFM and FM phases and these parameters will certainly affect the magnetization dynamics in a poorly controlled way.

The simple simulations of the behavior of two spins shown in the paper is meant for a clear and simple demonstration of the fact that a rapid change of the exchange interaction between two spins simply cannot lead to instantaneous magnetization dynamics and must show a latency. Our experiments do show the latency and we find that this is a very important message that must be communicated to the scientific community.

We strongly disagree with the referee’s opinion that our model does not support our findings too much. We model the behavior of two exchange coupled spins. When the sign of the exchange interaction changes instantaneously, despite the fast dynamics of $M(t)$, it first shows a latency and only afterwards the spins start to move in the exchange field. In this respect, the model does support our findings and the fact of the latency being present in both experiments and simulations is beyond any reasonable doubt. The slope $dM(t)/dt$ after the latency is indeed different in the simulations and the experiment, but this is certainly no surprise. In our simple two-spin model, after the sign change of the exchange interaction and after the latency, $dM(t)/dt$ is defined by the strength of the effective field of the exchange interaction, which is of the order of $10^2 - 10^3$ Tesla, much larger than the applied magnetic field. In real samples, the appearance of FM nuclei with emerging magnetizations in random directions makes the dynamics more complex. A more realistic model of FeRh should therefore consist of many of these antiparallel spins, which after the (here) demonstrated period of latency nucleate into many ferromagnetic domains that will subsequently grow. However, such a study of moving domain walls is a complete project of its own, and taking all these details into consideration here would

only drift away from the main message of our work which comprises the presence of latency.

10. Why would the fundamental delay of the phase transition be so different in the Pt vs. the Au sample? I think there is a significant difference between the two experimentally obtained values? The presented theory does not take the cap layer into account at all.

10) To understand the origin of the difference in the latency for FeRh/Au and FeRh/Pt, one has to be able to experimentally distinguish exchange-driven spin dynamics from the dynamics of the exchange interaction. However, this is not yet possible and the difference remains unclear to us. Moreover, our model only takes into account the spin systems and lattice expansion of FeRh. We did not include any capping layers.

So what do we learn from the FeRh/Au results? The difference in the latency for Pt and Au capping is significant and there is no discussion of this at all – it is just noted, that the latency is of “same order of magnitude”. The author should discuss this issue in the manuscript.

- **Response:** From the FeRh/Au results we do learn that the latency is present also in this sample, thus being a very general effect. This fact together with experiments and simulations on FeRh/Pt all strengthen the main message of the paper – the presence of the latency.

The magnetization dynamics in realistic samples would depend on the magnetic anisotropy in both the AFM and FM phases. At this moment there are no reports about the effects of Pt and Au layers on the magnetic anisotropy of FeRh, but this effect must be certainly present [J. Okabayashi *et al.*, *Phys. Rev. B* **103** (2021)], [C.-J. Lin *et al.*, *J. Magn. Magn. Mater.* **93** (1991)]. Moreover, there is hardly any data on magnetic anisotropy in the AFM phase and effects of capping layers on it. We have added a comment explaining the questions raised by the experiments on FeRh/Au in the revised lines [295 - 302].

“Since the FeRh/Pt and FeRh/Au interfaces must result in different anisotropies [33], [34] and magnetization dynamics also depends on magnetic anisotropy for both the antiferromagnetic and ferromagnetic phases, it would be interesting to see how the kinetics of the magnetic phase transition is affected by differences in capping layer. The latency for FeRh/Au, where the THz emission is dominated by magnetic dipole sources, is shown in the inset of Fig. 7(b). The extracted latency is slightly higher compared to FeRh/Pt. However, the signal-to-noise ratio was much worse which resulted in a higher estimate due to a higher normalized noise level.”

In Figure 3: The electric field strength in (b) at 120 mT is smaller than in (a) for 105 mT. This is not consistent and must be explained.

The two datasets of Fig. 3(a) and 3(b) were taken weeks apart. Small day-to-day fluctuations in the alignment in the setup resulted in a lower ΔE peak at the highest magnetic field strength.

The normalization of the field strength in Fig. 3 covers the issue with the day-to-day fluctuations, but it also misses the H-field dependence of the THz field amplitude as this is related to better alignment of the FM domains along the applied H-field. I would prefer the former version of the figure with a short statement about amplitudes in the methods section.

- **Response:** We will use the previous version of Fig. 3 and added a statement in the methods section in lines [464 - 467].

“Day-to-day fluctuations in the setup resulted in differences in the detected ΔE . This resulted in a lower ΔE peak signal in Fig. 3(b) while having a higher applied magnetic field compared to the ΔE signal in Fig. 3(a).”

L215 & Figure 5: The authors state that their fit function results in a large mismatch with the experimental data. This is obvious, as they do not include the delay of the exponential rise. Please fit the data with a double exponential fit function that includes a delay. Otherwise the inset in Fig. 5 (a) is also rather meaningless. Please also scale the inset to maximize the view on the presented data.

In (b) one should not fit a linear function in order to determine the delay of the onset. Please use again a step- or error-function in combination with a linear function instead.

We want to emphasize that the fit was first done with the understanding that there were only two regimes of dynamics, the rise and relaxation. The mechanism of the latency is still not understood well enough to simply fit it with some modified function. Modifying the function without a good physical reason would also change the other extracted fit parameters that will put doubt in the overall explanation of the fit.

I disagree here! Adding a delay to the fit function is obvious and is a first order approximation for any physical mechanism which causes this latency. This is a clear experimental finding and the reasoning is done in a second step.

I think it is not appropriate to fit data with an obviously wrong model just to show that it is wrong, when a rather simple analytical model is at hand.

At the same time, including the delay into the fits, would give a more objective value of the latency time. As seen from Fig. 5b, it seems that the determined latency of 20ps is not a result of a fit but read by eye and is kept constant for different magnetic fields.

I think that this value must be determined in a more objective manner. The same applies for the delay of the FeRh/Au sample as shown in Fig. 7.

- **Response:** In response to the referee’s comment, we fitted the data with a function which was inspired by the Johnson-Mehl-Avrami-Kolmogorov equation. The

discussion is added in lines [255 - 267]. And lines [294 - 301]. We also refer to comment (3) of referee 1 where we also discuss this issue.

Figure 7: please bin the data and add error bars instead of cluttering the complete plot with data points.

Again the fits in (a) and (b) do not look very convincing to me.

We have binned the data in Fig. 7 and removed the fits in Fig. 7(a).

How about error bars?

- **Response:** Figure 7 now includes error bars.

Figure 9: Can one neglect heat diffusion out of the phononic system of FeRh into the Pt cap and MgO substrate layer? The FM saturation is reached already after 30 ps in the simulation, while the experiment needs about 300 ps, can you discuss the discrepancy? Why does the FM saturation moment not depend on the external magnetic field as in the experiment. Moreover, it is only 0.8 μB and not the full 3.8 μB of Fe. I think the 3.8 μB for Fe is also a rather large value.

Please also shift the time zero from 3 ps to 0 ps in accordance to the experiment.

Accordingly, the onset at 10 ps is only 7 ps after the first pump pulse in the simulation and not approx. 10 ps!

I would also appreciate the usage of experimental SI units even for the simulations.

We focused in our model on the qualitative description of the transition to the ferromagnetic state, in particular the latency behavior. Therefore, we did not include any heat diffusion processes. Moreover, we consider a homogeneous heating across the whole sample, which is justified by the fact that the film is only 40 nm. This results in the ferromagnetic saturation and discrepancy in the timescale seen in the simulation. In our revised version, we have simplified our model and normalized the saturation moment of Fe for clarity.

Normalizing the magnetization does not resolve the discrepancy between your model and the general theoretical expectations regarding the magnetic moment. I am still missing a discussion about the discrepancy of FM saturation times between the model (30ps) and the experiment (300ps). I guess there are good reasons for that, but it needs to be discussed in the manuscript.

- **Response:** We reemphasize that the main message of our simple model is that an ultrafast reorientation to the ferromagnetic spin order is not possible even under and instantaneous change in the exchange integral. The large discrepancy between our model (30 ps) and experiment (300 ps) can be due to the various approximations and physical effects that are left out for the sake of simplicity, including the heat diffusion out of the system. As we have already mentioned in our response to

comment (7), one should consider a many spin model in order to include nucleation and growth kinetics. We have added comments on this discrepancy in our discussion in lines [385 - 390].

“Note that our model only considers two exchange coupled macrospins without any heat dissipation out of FeRh. The spin dynamics driven by the exchange is inherently fast, resulting in a large discrepancy between the simulation and the experimental data. However, despite the oversimplification, a latency period before the growth of magnetization is present nonetheless. This shows that this period right after the laser-excitation cannot be ignored.”

L395: the second pump pulse should be much smaller than the first one, in order to probe a homogeneously excited region with no in-plane gradients.

We chose to keep the beamspot equally large to maximize the signal-to-noise ratio. Moreover, a smaller spot of second pump pulse did not affect the overall dynamics.

Here I am a bit puzzled, as I would strongly expect a change of the FM response for different pump2 spot sizes as this is essentially a variation in the pump fluence. Did the authors measure any fluence dependence at all?

- **Response:** We performed the fluence dependence for both pumps. Only varying the fluence of pump 1 affects the FM response, this behavior is similar to the one reported earlier by Radu et al. [Radu et al., Phys. Rev. B, **81**, 104415 (2010)]. Our minimum threshold required to drive the phase transition is 4 mJ cm^{-2} , this corresponds to the same threshold reported by Radu et al. Of course, the THz emission signal is linearly dependent on the fluence pump 2, with the intercept at zero. This shows that our fluence of the second pump is far below saturation [J.P. Ferrolino et al., J. Phys: Conf. Ser. **1943**, 012035 (2021)]. Moreover, these measurements at lower fluences of pump 2 give the very same dynamics with a worse signal-to-noise ratio (see supplementary Fig. S7).

Figure S7 The double pump THz emission signal where (a) the fluence of pump 1 is varied while pump 2 is fixed or (c) where the fluence of pump 1 is fixed while pump 2 is varied. The ΔE signal of (a) was fitted with the function $A_1 \left(1 - e^{-\frac{t}{\tau_{Rise}}}\right) + A_2 e^{-\frac{t}{\tau_{Relax}}}$ and the extracted τ_{Rise} and τ_{Relax} as a function of the pump 1 fluence is shown in (b). (d) The maximum ΔE signal as a function of the fluence of the pump 1 (black) or pump 2 (red). The measurement was done by varying the fluence of one pump while the fluence of the other pump was fixed at the maximum fluence.

Reviewers' Comments:

Reviewer #1:

Remarks to the Author:

In the latest revision the authors have made a number of additional changes and also responded to the comments of myself and another referee. The results are I think quite interesting and useful input to the community, in that it shows another experimental confirmation that the magnetic component of the ultrafast phase transition in FeRh is indeed significantly delayed using a new experimental technique, agreeing with some earlier results using time-resolved MOKE (ref. 9) that were careful to avoid artefacts that likely affected previous similar studies. The fact that the delay is confirmed using a completely independent technique should largely settle any lingering doubts.

I am at this point willing to recommend publication, contingent on a few changes:

- 1) The authors need to revise their summary to not claim that the latency is "previously overlooked". The delay in the inset of FM order was in fact clearly identified already in ref. 9. This needlessly obscures the real contribution, which is to show that using a new method less prone to artefacts that this latency is confirmed, and may be a more general feature of 1st order magnetic transitions and may not just be from a structural bottleneck. Note I say "may" since the authors' experiments do not actually show that, but their discussion and toy model suggest this.
- 2) The authors should at least remark on the discrepancies between their model and the data. I would not expect that the model is anything but qualitative, but the authors should not just ignore the fact that the model predicts different onset times for different fluences, while the experiment does not. Also, the authors should use the same fit function (eq 2) to extract latency time from their model, and compare this to the experiment.
- 3) Equation (1) and the surrounding discussion is not adequately explained. The procedure that the authors are using to normalize the data is unclear, and it would appear to still rely on many unstated assumptions (hard to say since Eq 1 is not derived, but simply presented without clear definitions of variables). Also, the claim that the time-traces of the THz emission are unchanged on spatial inversion is only qualitatively true. Since this whole process does not appear to be central to the main argument of the paper, I would at this point recommend simply dropping the attempt to quantitatively separate magnetic dipole and electric dipole effects. Thus I would remove the text from lines 199-236, and just talk more qualitatively about the fact that both ED and MD contributions are important. This would also require removing panel b from fig 4.
- 4) In line 305 the authors claim that the FeRh/Au data presented in Fig. 7b (inset) "suggest the presence of a latency regime nonetheless". This is misleading. The inset in fact shows that the latency is, if present, is not possible to quantify to a level exceeding experimental uncertainty. This is not the same as the conclusion the authors draw, but it is the only one supported by the data.
- 5) The sentence from lines 318-321, "The strongly varying THz sources..." is unclear and draws a conclusion that is not sufficiently justified by the presented arguments. This needs to be revised for clarity.

Reviewer #2:

Remarks to the Author:

In my opinion, the authors have made great improvements to the manuscript during the review process and have addressed nearly all issues which have been raised.

I recommend publication in NatComm

Referee #1

The Referee's comments are in black.

Our responses to the Referee are in blue.

In the latest revision the authors have made a number of additional changes and also responded to the comments of myself and another Referee. The results are I think quite interesting and useful input to the community, in that it shows another experimental confirmation that the magnetic component of the ultrafast phase transition in FeRh is indeed significantly delayed using a new experimental technique, agreeing with some earlier results using time-resolved MOKE (ref. 9) that were careful to avoid artefacts that likely affected previous similar studies. The fact that the delay is confirmed using a completely independent technique should largely settle any lingering doubts. I am at this point willing to recommend publication, contingent on a few changes:

- 0) We thank the Referee for the encouraging words and their critical look which helped us in further improving our manuscript. Below you will find the point-by-point responses to the comments.
- 1) The authors need to revise their summary to not claim that the latency is “previously overlooked”. The delay in the inset of FM order was in fact clearly identified already in ref. 9. This needlessly obscures the real contribution, which is to show that using a new method less prone to artefacts that this latency is confirmed, and may be a more general feature of 1st order magnetic transitions and may not just be from a structural bottleneck. Note I say “may” since the authors’ experiments do not actually show that, but their discussion and toy model suggest this.
- 1) We thank the Referee for this suggestion. Following the request, we have revised the summary accordingly. In particular, we have removed the sentence about “previously overlooked” latency and insert new sentences stating – “Therefore, our work reveals that the latency, previously identified in Ref. [9], must be a general feature of first order magnetic phase transitions and not necessarily unique to FeRh or even to a particular FeRh film”. See lines [371 - 373]
- 2) The authors should at least remark on the discrepancies between their model and the data. I would not expect that the model is anything but qualitative, but the authors should not just ignore the fact that the model predicts different onset times for different fluences, while the experiment does not. Also, the authors should use the same fit function (eq 2) to extract latency time from their model, and compare this to the experiment.
- 2) We have added an additional remark to our model in lines [352 - 361].

“Note that the slope $d\mathbf{M}(t)/dt$ after the latency is indeed different in the simulations compared with the experiment, but this is certainly not surprising. In our simple two-spin model, after the sign change of the exchange interaction and after the latency, $d\mathbf{M}(t)/dt$ is defined by the strength of the effective field of the exchange interaction. This is of the order of $10^2 - 10^3$

Tesla. In real samples, the appearance of ferromagnetic nuclei with emerging magnetizations in random directions makes the dynamics more complex. A more realistic model of FeRh should therefore consist of many of these antiparallel spins, which after a latency period will nucleate into many ferromagnetic domains that will subsequently grow. However, such a computational study, including moving domain walls, is presently a challenge and certainly a complete project of its own.”

In lines [333 - 338] we add:

“Both experimental and computational results reveal a latency period in the growth of net magnetization. Fitting the simulated data for the case when the exchange constant J_{eff} changes sign instantaneously at $\mu_0 H = 105$ mT with our modified Johnson-Mehl-Avrami-Kolmogorov equation, we found a latency of 9.33 ± 0.01 ps, which is in good agreement with the experimentally observed latency”

- 3) Equation (1) and the surrounding discussion is not adequately explained. The procedure that the authors are using to normalize the data is unclear, and it would appear to still rely on many unstated assumptions (hard to say since Eq 1 is not derived, but simply presented without clear definitions of variables). Also, the claim that the time-traces of the THz emission are unchanged on spatial inversion is only qualitatively true. Since this whole process does not appear to be central to the main argument of the paper, I would at this point recommend simply dropping the attempt to quantitatively separate magnetic dipole and electric dipole effects. Thus I would remove the text from lines 199-236, and just talk more qualitatively about the fact that both ED and MD contributions are important. This would also require removing panel b from fig 4.

- 3) We followed the advice of the Referee and removed the discussion of the MD and the ED sources together with Fig. 4(b) from the main text. We revised this paragraph to a qualitative discussion on the ED and MD contributions, see lines [193 – 199]

“This fundamental difference in the behavior of the sources under space inversion can be employed to retrieve the dynamics of the magnetic (E^{MD}) and electric (E^{ED}) dipole contributions to the laser-induced changes in the THz emission.

The dipole contributions to the ΔE signal pumped from the MgO-side are expressed as $\Delta E^{\text{MgO}} = E^{\text{MD}} + E^{\text{ED}}$ and from the Pt-side the signal is $\Delta E^{\text{Pt}} = E^{\text{MD}} - E^{\text{ED}}$. Hence in order to obtain the best signal-to-noise ratio, one has to perform the measurements pumping from the MgO-side, when the two sources interfere constructively.”

- 4) In line 305 the authors claim that the FeRh/Au data presented in Fig. 7b (inset) “suggest the presence of a latency regime nonetheless”. This is misleading. The inset in fact shows that the latency is, if present, is not possible to quantify to a level exceeding experimental uncertainty. This is not

the same as the conclusion the authors draw, but it is the only one supported by the data.

- 4) We have revised our conclusion on the data on FeRh/Au in lines [267 - 269].

“While the data do not contradict the possible presence of a latency, the experimental uncertainty hampers quantitative estimates of the latter.”

- 5) The sentence from lines 318-321, “The strongly varying THz sources...” is unclear and draws a conclusion that is not sufficiently justified by the presented arguments. This needs to be revised for clarity.

- 5) We have followed the advice of the Referee and revised the sentences taking into account that the discussion of the MD and ED sources has now take less attention in the paper (see issue 3 of the report).

We have removed the criticized sentence: “The strongly varying THz sources between MgO/FeRh/Pt and MgO/FeRh/Au rules out any interferences between the electric fields of the electric- and magnetic-dipole sources as a cause of the absence of any sub-picosecond dynamics in the double-pump $\Delta E(\mu_0 H) - \Delta E(15, 30 \text{ mT})$ signal.”

The conclusion made in the next sentence is now supported by earlier paper Ref. 12. Instead of “On this basis, ...” we now write “As explained in Ref. 12, ...”. See line [282].

Referee #2

The Referee’s comments are in black.

Our responses to the Referee are in blue.

In my opinion, the authors have made great improvements to the manuscript during the review process and have addressed nearly all issues which have been raised. I recommend publication in NatComm.

- 0) We would like thank the Referee for the kind words and we appreciate the critical comments which helped us to improve the quality of our manuscript.

Reviewers' Comments:

Reviewer #1:

Remarks to the Author:

I am largely satisfied with the revisions made in this last round, as most of my concerns were addressed. There is only one small point: in the summary paragraph (lines 374-387) the authors overstate one aspect of their conclusions. Although in the main text they are careful to say that within the framework of their simplified model the latency is a general property of first order transitions, here in the summary this becomes a statement that the work has in general shown this to be true, without qualification. Since the authors themselves note the shortcomings of their model I do not see this as justified, and a casual reader might jump immediately to this paragraph and be misled. I would insist that the authors revise the summary to make it clear that the latency time is a general feature of magnetic phase transitions only insofar as a simple model of the process suggests, but that more work would need to be done on the model and with other systems to establish true generality.

Referee #1

The Referee's comments are in black.

Our responses to the Referee are in blue.

- 1) I am largely satisfied with the revisions made in this last round, as most of my concerns were addressed. There is only one small point: in the summary paragraph (lines 374-387) the authors overstate one aspect of their conclusions. Although in the main text they are careful to say that within the framework of their simplified model the latency is a general property of first order transitions, here in the summary this becomes a statement that the work has in general shown this to be true, without qualification. Since the authors themselves note the shortcomings of their model I do not see this as justified, and a casual reader might jump immediately to this paragraph and be misled. I would insist that the authors revise the summary to make it clear that the latency time is a general feature of magnetic phase transitions only insofar as a simple model of the process suggests, but that more work would need to be done on the model and with other systems to establish true generality.
- 1) We thank the referee for carefully reading our manuscript and agree with his or her point. We have rephrased our summary paragraph such that we do not overstate our conclusions.

“Therefore, our work strongly suggests that, within the framework of our simple model, the latency, previously identified in Ref. [9], must be a general feature of first order magnetic phase transitions and not necessarily unique to FeRh or even to a particular FeRh film. At the same time, it is also clear that more work would need to be done on the model for FeRh and other similar systems to establish true generality”